



# Analytic characterization of random errors in spectral dual-polarized cloud radar observations

Alexander Myagkov[1] and Davide Ori[2]

[1]Radiometer Physics GmbH, Meckenheim, Germany
[2]Institute for Geophysics and Meteorology, University of Cologne, Cologne, Germany

**Correspondence:** Alexander Myagkov (alexander.myagkov@radiometer-physics.de)

**Abstract.** This study presents the first-ever complete characterization of random errors in dual-polarimetric spectral observations of meteorological targets by cloud radars. The characterization is given by means of mathematical equations for joint probability density functions (PDF) and error covariance matrices. The derived equations are checked for consistency using real radar measurements. One of the main conclusions of the study is that the convenient representation of spectral polari-

metric measurements including differential reflectivity $Z_{DR}$, correlation coefficient $\rho_{HV}$, and differential phase $\Phi_{DP}$ is not suited for the proper characterization of the error covariance matrix. This is because the aforementioned quantities are complex, non-linear functions of the radar raw data and thus their error covariance matrix is commonly derived using simplified linear relations and by neglecting the correlation of errors. This study formulates the spectral polarimetric measurements in terms of a different set of quantities that allows for a proper analytic treatment of their error covariance matrix. The results given in this

study allow for utilization of spectral polarimetric measurements for advanced meteorological applications, among which are variational retrieval techniques, data assimilation, and sensitivity analysis.

## 1 Introduction

Cloud radars are a major component of state-of-the-art, ground-based observation platforms (Illingworth et al., 2007; Kollias et al., 2020). Their unique capabilities make these instruments extremely valuable for cloud and precipitation research. First,

these radars have Doppler capabilities, i.e. can independently characterize hydrometeors coexisting in the same volume but moving with different speeds relative to the radar (Kollias et al., 2007). Second, the high sensitivity and vast dynamic range make cloud radars capable of measuring return signals from a wide range of particles sizes, which is a challanging task for other instruments like lidars (Bühl et al., 2013). Third, due to relatively low attenuation of microwave signals by liquid water, cloud radars profile clouds up to the top even in presence of light-to-moderate rain. These capabilities promote cloud radars for

investigation of different formation and development processes throughout the lifecycle of clouds. For instance, cloud radars help to characterize initial ice formation and development in mixed-phase clouds (Bühl et al., 2019), improve characterization of pure liquid clouds (Rusli et al., 2017; Acquistapace et al., 2017), estimate rates of aggregation (Kneifel et al., 2015, 2016) and riming (Kalesse et al., 2015; Moisseev et al., 2017; Kneifel and Moisseev, 2020), and quantitatively analyse solid and liquid precipitation (Matrosov, 2005; Matrosov et al., 2006, 2008; Tridon and Battaglia, 2015; Tridon et al., 2017, 2019).



Many cloud radars have dual-polarization capabilities. An interest in polarimetry-based methods in the cloud radar community has been growing, which is indicated by a number of studies during the last decade (Matrosov et al., 2012; Oue et al., 2015; Lu et al., 2015; Myagkov et al., 2016a, b; Matrosov et al., 2017; Oue et al., 2018; Myagkov et al., 2020). Vertically pointed cloud radars often operate in the LDR-mode (Linear Depolarization Ratio), i.e transmit a linearly-polarized wave (either horizontally or vertically) and receive co- and cross-polarized components of the backscattered signal (e.g Görsdorf et al., 2015).

The LDR-mode is efficient for clutter removing and detection of the melting layer and columnar-shaped ice particles. As shown by Matrosov et al. (2001), however, the applicability of the LDR mode at low elevation angles might be limited due to its high sensitivity to the orientation of cloud particles. Therefore, scanning polarimetric cloud radars often have polarimetric modes which are less sensitive to the orientation. One of such modes is the hybrid mode (also denoted as the STSR (Simultaneous Transmittion and Simultaneous Reception) or STAR (Simulteneous Transmittion And Reception) mode in literature). Radars

with the hybrid mode emit the horizontal and vertical components of the transmitted wave simultaneously (Myagkov et al., 2015; Bringi and Chandrasekar, 2001, Sec. 4.7). Cloud radars with the hybrid mode allow for adoption of polarimetry-based methods having been developed during last several decades for centimetre-wavelength meteorological radars (further denoted as precipitation radars).

    Operational precipitation radars are used by weather services to continuously scan the atmosphere providing polarimetric

variables integrated for a scattering volume. In addition to the integrated quantities, cloud radars with the hybrid mode enable spectrally-resolved polarimetric observations and, therefore, can provide the same set of polarimetric variables for different types of cloud particles coexisting in the same resolution volume (Oue et al., 2015; Myagkov et al., 2016b, 2020). Spectral observations are in general possible with precipitation radars (Spek et al., 2008; Dufournet and Russchenberg, 2011; Pfitzenmaier et al., 2018). Such measurements, however, are not performed by operational radars due to fast azimuth scanning.

Spectral polarimetry can be used for a development of advanced retrieval methods. For example variational retrievals developed for dual-frequency spectra (Tridon and Battaglia, 2015; Tridon et al., 2017) could be applied also to spectral polarimetry. Moisseev and Chandrasekar (2007) presented first attempts to retrieve profiles of raindrop-size distributions using polarimetric spectra from a precipitation radar. This approach, however, has not been yet explored in polarimetric cloud radars.

    Recent review studies (Zhang et al., 2019; Morrison et al., 2020; Ryzhkov et al., 2020) demonstrate that polarimetric ob-

servations from precipitation radar networks are highly beneficial for the evaluation and development of numerical weather prediction and cloud resolving models. The high value of polarimetric observations is given by their sensitivity to microphysical properties of cloud and precipitation particles such as size, shape, number concentration, state of matter, density, and orientation (Kumjian, 2013). Polarimetric cloud radars are not yet widely used for model improvement. This, however, does not indicate that cloud radar polarimetry is not informative relative to precipitation radars. Conversely, the cloud radar spectral

polarimetry can essentially complement available measurements.

    The development of both quantitative retrievals and data assimilation algorithms requires the characterization of the systematic and random measurement errors. The former type of errors is solved by a calibration. Calibration aspects of polarimetric quantities have been intensively studied for both precipitation and cloud radars (Chandrasekar et al., 2015) and are out of the scope of this study. In the case of radar observations of meteorological targets, random errors can be characterized from





measurements if raw (unaveraged) data are available. Cloud radars, however, rarely store raw data because of high data rate. Therefore, commonly used approaches to characterize random errors are based on statistical models of the received radar signals. Random errors of radar signals can be represented by a joint probability density function (PDF) of amplitudes and phases in the two orthogonal polarimetric channels. The joint PDF for polarimetric observations obtained for a single pulse can be found in Middleton (1996, chapter 9.2). Single-pulse measurements, however, are rarely used in the radar meteorology

because of the low sensitivity and higher requirement for storage space. The observed radar spectra, almost always, result from the averaging of a number of return pulses. Unfortunately, a solution for the case of averaging over a number of pulses is not yet available in literature.

A number of studies (e.g. Hogan (2007); Cao et al. (2013); Yoshikawa et al. (2014); Chang et al. (2016); Huang et al. (2020)) characterize the joint PDF of polarimetric radar measurements by the error covariance matrix. There are, however, problems

with existing approximations of the error covariance matrix for polarimetric observations. First, the elements in the main diagonal of the error covariance matrix – variances of random errors – are found using the first-order Taylor approximation following Bringi and Chandrasekar (2001). Conventional polarimetric variables such as differential reflectivity, correlation coefficient, and differential phase are , however, highly non-linear functions. Therefore, the approximation may lead to biases in the error variance estimates especially when signal-to-noise ratios (SNR) and/or the number of averaged samples is low.

This problem becomes important for cloud radars collecting polarimetric variables with a high spatial, temporal, and spectral resolution. Second, non-diagonal components of the error covariance matrix are typically set to zero assuming no correlation between errors in measured quantities but validity and effects of this assumption are not discussed. The information content of measurements is, however, higher when errors are correlated (chapter 3.2.6 in Rodgers, 2000) and therefore, non-negligible off-diagonal elements of the covariance matrix should not be ignored.

This study will review the measurement method of spectral polarimetry with radars operating in the hybrid mode in Sec. 2. In Sec. 3 the likelihood functions of the common polarimetric radar variables are rigorously derived. The error covariance matrix of polarimetric measurements is derived in Sec. 4 by taking into account the correlations among the various measurement random errors. In Sec. 5 the validity of expressions derived for the likelihood functions and error covariance matrix is checked using real raw measurements from a cloud radar.

## 85   2   Spectral polarimetry in the hybrid mode

Radar polarimetric measurements are made in an orthogonal measurement basis defined by feeders of the antenna system. In the hybrid mode the measurement basis is typically Cartesian and formed by the horizontal ($h$) and vertical ($v$) components. Further this basis is denoted as the $h$–$v$ basis. Dual-polarimetric cloud radars have two receivers dedicated to the orthogonal polarimetric components of the received signal. For each transmitted pulse (the term pulse is used throughout the study, al-

though for radars with frequency modulated continuous wave signals a chirp would have been implied) the receivers provide range profiles of in-phase $I_{h,v}$ and quadrature $Q_{h,v}$ components, where indices $h$ and $v$ denote the polarization state. Note, that this study does not cover the radar signal processing to get the $I_{h,v}$ and $Q_{h,v}$ profiles. This information can be found in a





radar handbook e.g. Skolnik (2008, Chapter 6). Using $N_{\text{fft}}$ profiles of $I_h + iQ_h$ and $I_v + iQ_v$, where $i$ is the imaginary unit, the radar calculates complex Doppler spectra in the horizontal and vertical channel, respectively, applying the Fast Fourier

Transformation (FFT) along the time dimension. The complex Doppler spectra are represented by complex amplitudes $\dot{S}$ for each spectral component and each range bin.

Different range bins as well as different spectral components are often considered to be statistically independent, because the corresponding complex amplitudes result from non-coherent scattering of numerous independently moving particles. Some correlation, however, can be expected due to sampling effects and the FFT spectral leakages (e.g. Sec. 5.3 in Marple, 2019). For

instance, the power scattered from particles located close to the end of a range bin is distributed between this and the following range bins. These effects depend on filter properties and used FFT windows. It is challenging to give a general analytical solution taking these effects into account. Therefore, these effects are out of the scope of this study. For the sake of simplicity the following analysis is shown only for a single range bin and a single spectral component. Since movements of particles in neighboring range and spectral bins are not related, statistical properties of an individual bin considered in the following

are not affected by sampling effects and spectral leakages. The neglection of the dependence of the neighboring bins leads to an underestimation of the information entropy when a complete spectrum and/or spectral profile is analyzed. This worst case assumption, however, allows for a relatively easy and universal characterization of measurement errors. Future studies may improve the error characterization by considering the sampling and leakage effects.

In the following, $\dot{S}_h$ and $\dot{S}_v$ denote the measured complex amplitudes of the analysed spectral component in the horizontal

and vertical channels, respectively (the dot hereafter denotes a complex quantity). Introduce a measurement column-vector

$$\hat{\boldsymbol{m}} = [\hat{R}_h, \hat{J}_h, \hat{R}_v, \hat{J}_v]^{\text{T}} \tag{1}$$

with $\hat{R}$ and $\hat{J}$ being real and imaginary parts of a complex amplitude $\dot{S}$, indices $h$ and $v$ denote the polarization state, $^{\text{T}}$ is the transposition sign, the overhat hereafter is used to emphasize measured quantities. The probability density function (PDF) of $\hat{\boldsymbol{m}}$, given the true covariance matrix $\boldsymbol{\Sigma}_m$ of $\hat{\boldsymbol{m}}$, can be written as follows:

$$f_m\left(\hat{\boldsymbol{m}}|\boldsymbol{\Sigma}_m\right) = (2\pi)^{-2} \det(\boldsymbol{\Sigma}_m)^{-\frac{1}{2}} e^{-\frac{1}{2}\boldsymbol{m}^{\text{T}}\boldsymbol{\Sigma_m}\boldsymbol{m}}. \tag{2}$$

Note that throughout the study a PDF is a function of measured quantities (e.g. $\hat{\boldsymbol{m}}$ in Eq. 2) with fixed parameters (e.g. $\boldsymbol{\Sigma}_m$ in Eq. 2). The same PDF is called a likelihood function if the measured quantities are fixed and the PDF is viewed as a function of parameters.

Doviak et al. (1979) showed that for meteorological targets $I$ and $Q$ components are jointly normal with zero mean, zero

correlation, and equal standard deviation. The authors explain that these properties are due to scattering from a large number of particles moving in an unpredictable way in a scattering volume. Since $N_{\text{fft}}$ is much smaller than the number of particles in a resolution volume, the properties are also valid for relations between $\hat{R}_h$ and $\hat{J}_h$ and between $\hat{R}_v$ and $\hat{J}_v$. The measured complex amplitudes $\dot{S}_h$ and $\dot{S}_v$, however, can be correlated. Taking these properties into account, the true covariance matrix





$\mathbf{\Sigma}_m$ is defined in the following way (Eq. 5.178 in Bringi and Chandrasekar (2001)):

$$\Sigma_m = \begin{pmatrix} \sigma_h^2 & 0 & q\sigma_h\sigma_v & s\sigma_h\sigma_v \\ 0 & \sigma_h^2 & -s\sigma_h\sigma_v & q\sigma_h\sigma_v \\ q\sigma_h\sigma_v & -s\sigma_h\sigma_v & \sigma_v^2 & 0 \\ s\sigma_h\sigma_v & q\sigma_h\sigma_v & 0 & \sigma_{v,}^2 \end{pmatrix},$$

(3)

where $\sigma_h$ is the standard deviation of $\hat{R}_h$ and $\hat{J}_h$, $\sigma_v$ is the standard deviation of $\hat{R}_v$ and $\hat{J}_v$, $q$ is the correlation between $\hat{R}_h$ and $\hat{R}_v$, and $s$ is the correlation between $\hat{R}_h$ and $\hat{J}_v$.

Since for meteorological targets $\hat{R}_h$ is not correlated with $\hat{J}_h$ and $\hat{R}_v$ is not correlated with $\hat{J}_v$, the absolute phases of $\dot{S}_h$ and $\dot{S}_v$ are uniformly distributed from 0 to $2\pi$ and, thus, uninformative. Therefore, the polarimetric observations in the hybrid mode can be represented by a $2 \times 2$ covariance matrix $\mathbf{B}$ (Eq. 4.130 in Bringi and Chandrasekar (2001)) instead of the true covariance matrix $\mathbf{\Sigma}_m$:

$$\mathbf{B} = \overline{\boldsymbol{ee}^{\mathrm{T}}} = \begin{pmatrix} B_{hh} & \dot{B}_{hv} \\ \dot{B}_{hv}^* & B_{vv} \end{pmatrix},$$

(4)

where

$$\boldsymbol{e} = (\dot{S}_h, \dot{S}_v)^{\mathrm{T}};$$

(5)

the overline indicates the expected value, $B_{hh}$ and $B_{vv}$ have meaning of total powers of the horizontal and vertical components of the received signal, respectively, $\dot{B}_{hv}$ is the covariance between the horizontal and vertical components of the received signal, and $*$ is the complex conjugation sign. Recall, that in this study the covariance matrix $\mathbf{B}$ corresponds to a single spectral component. Such spectral representation of vector signals was introduced by Wiener (1930).

The elements of $\mathbf{B}$ are related to the statistics of the complex amplitudes $\dot{S}_h$ and $\dot{S}_v$ as follows:

$$B_{hh} = \mathrm{var}(\hat{R}_h) + \mathrm{var}(\hat{J}_h) = 2\sigma_h^2,$$

(6)

$$B_{vv} = \mathrm{var}(\hat{R}_v) + \mathrm{var}(\hat{J}_v) = 2\sigma_v^2,$$

(7)

$$\dot{B}_{hv} = R_{hv} + iJ_{hv} = (q + js)\sigma_h\sigma_v,$$

(8)

where $R_{hv}$ and $J_{hv}$ are real and imaginary parts of $\dot{B}_{hv}$.

In the precipitation radar community, dual-polarized measurements are rarely represented by $\mathbf{B}$. Instead a set of polarimetric variables is used. Therefore, the same polarimetric variables (but spectrally resolved) are introduced in this study. Introduce a vector

$$\boldsymbol{c} = (B_{hh}, Z_{DR}, \rho_{HV}, \Phi_{DP})^T,$$

(9)





where $Z_{DR}$ is the differential reflectivity, $\rho_{HV}$ is the correlation coefficient, and $\Phi_{DP}$ is the differential phase. In this study $Z_{DR}$, $\rho_{HV}$, and $\Phi_{DP}$ are defined for each spectral line using elements of corresponding **B**:

$$Z_{DR} = \frac{B_{hh}}{B_{vv}}, \tag{10}$$

$$\rho_{HV} = \sqrt{\frac{R_{hv}^2 + J_{hv}^2}{B_{hh}B_{vv}}}, \tag{11}$$

$$\Phi_{DP} = \mathrm{atan}\left(-\frac{J_{hv}}{R_{hv}}\right). \tag{12}$$

## 3 Likelihood of elements of the covariance matrix B

Assume the following problem. The state of the atmosphere is represented by the state vector $\boldsymbol{x}$. A forward model $F$ maps $\boldsymbol{x}$ into a vector

$$F(\boldsymbol{x}) = \boldsymbol{b} = (B_{hh}, R_{hv}, J_{hv}, B_{vv})^{\mathrm{T}} \tag{13}$$

in the space of observations. The actual measurement vector is

$$\hat{\boldsymbol{b}} = (\hat{B}_{hh}, \hat{R}_{hv}, \hat{J}_{hv}, \hat{B}_{vv})^{\mathrm{T}} = \boldsymbol{b} + \boldsymbol{\epsilon}, \tag{14}$$

where

$$\hat{B}_{hh} = \left\langle \dot{S}_h \dot{S}_h^* \right\rangle, \tag{15}$$

$$\hat{R}_{hv} = \mathrm{Re}\left(\left\langle \dot{S}_h \dot{S}_v^* \right\rangle\right), \tag{16}$$

$$\hat{J}_{hv} = \mathrm{Im}\left(\left\langle \dot{S}_h \dot{S}_v^* \right\rangle\right), \tag{17}$$

$$\hat{B}_{vv} = \left\langle \dot{S}_v \dot{S}_v^* \right\rangle, \tag{18}$$

are constituents of the measured covariance matrix $\hat{\mathbf{B}}$ and $\boldsymbol{\epsilon}$ represents the vector of measurement random errors in each component of $\hat{\boldsymbol{b}}$. In Eqs. 15–18 $<>$ denotes averaging over $N_s$ complex spectra, Re and Im are the real and imaginary parts of a complex number. What is the likelihood of $\hat{\boldsymbol{b}}$ given the state vector $\boldsymbol{x}$? In the case the forward model provides a unique and accurate relation between $\boldsymbol{x}$ and $\boldsymbol{b}$, the problem is equivalent to finding $f_b(\hat{\boldsymbol{b}}|\boldsymbol{b}, N_s)$ – the likelihood of $\hat{\boldsymbol{b}}$ – given the true vector of measurements $\boldsymbol{b}$ and the number of averaged spectra $N_s$. The derivation of $f_b(\hat{\boldsymbol{b}}|\boldsymbol{b}, N_s)$ provided in this section includes several steps. In Sec. 3.1 the polarimetric basis is changed to cancel the correlations between the orthogonal components of the measured vector. In the new basis the likelihood function can be represented by a product of likelihood functions, each of which is a function of only a single independent element. In Sec. 3.2 a formal derivation of the likelihood function in this new basis is provided. The solution for $f_b(\hat{\boldsymbol{b}}|\boldsymbol{b}, N_s)$ is given in Sec. 3.3 converting back to the original space and applying the rule of change of variables. As it was mentioned above, the radar observations are often represented by the vector $\boldsymbol{c}$. Therefore, Sec. 3.3 also provides the likelihood $f_c(\hat{\boldsymbol{c}}|\boldsymbol{b}, N_s)$.





## 3.1 Diagonalization of the covariance matrix B

As it was previously mentioned, $\dot{S}_h$ and $\dot{S}_v$ are, in general, correlated. There is, however, always a basis, in which the projections of $\dot{S}_h$ and $\dot{S}_v$ become completely uncorrelated. This basis is further denoted as the $c$–$x$ (co-polar and cross-polar) basis. The conversion of the vector $e$ in the $h$–$v$ basis to the vector $e_D$ in $c$–$x$ basis is made using the unitary operator $\mathbf{Q}$:

$$e_D = \begin{pmatrix} \dot{S}_c \\ \dot{S}_x \end{pmatrix} = \mathbf{Q}e \tag{19}$$

The calculation of the matrix $\mathbf{Q}$ is given in Appendix A. Real and imaginary parts of $\dot{S}_c$ are jointly distributed normally with the zero mean, zero correlation, and standard deviation $\sigma_c$. Real and imaginary parts of $\dot{S}_x$ are also jointly distributed normally with zero mean, zero correlation, but have, in general, a different standard deviation $\sigma_x$.

The covariance matrix $\mathbf{D}$ of $e_D$ has the diagonal form and can be found as follows

$$\mathbf{D} = \begin{pmatrix} D_{cc} & 0 \\ 0 & D_{xx} \end{pmatrix} = \mathbf{Q}^\dagger \mathbf{B} \mathbf{Q}. \tag{20}$$

In Eq. 20 $\dagger$ is the Hermitian conjugate. The elements of the matrix $\mathbf{D}$ can be found as follows:

$$D_{cc} = q_{11}^2 B_{hh} + |\dot{q}_{12}|^2 B_{vv} - 2q_{11}\left(R_{12}R_{hv} + J_{12}J_{hv}\right) \tag{21}$$

$$D_{xx} = |\dot{q}_{12}|^2 B_{hh} + q_{11}^2 B_{vv} + 2q_{11}\left(R_{12}R_{hv} + J_{12}J_{hv}\right) \tag{22}$$

where $\dot{q}_{nm}$ are elements of $\mathbf{Q}$ with $n$ and $m$ being indices of row and column, respectively;

$$\dot{q}_{12} = R_{12} + iJ_{12}. \tag{23}$$

Similar to relations between the powers and the standard deviations given in Eqs. 6 and 44, $\sigma_1$ and $\sigma_2$ are related to $D_{cc}$ and $D_{xx}$, respectively:

$$D_{cc} = \mathrm{var}(R_c) + \mathrm{var}(J_c) = 2\sigma_c^2 \tag{24}$$

$$D_{xx} = \mathrm{var}(R_x) + \mathrm{var}(J_x) = 2\sigma_x^2 \tag{25}$$

The measured values $\hat{D}_{cc}$,

$$\hat{D}_{cx} = \hat{R}_{cx} + i\hat{J}_{cx}, \tag{26}$$

and $\hat{D}_{xx}$ represent elements of the matrix $\hat{\mathbf{D}}$:

$$\hat{\mathbf{D}} = \mathbf{Q}^\dagger \hat{\mathbf{B}} \mathbf{Q}. \tag{27}$$

Note, that the operator $\mathbf{Q}$ is the same as in Eq. 20 and not recalculated using $\hat{\mathbf{B}}$.





## 3.2 Likelihood function in the c–x basis

By definition, the off-diagonal elements of the covariance matrix $\mathbf{D}$ are zeros (see Eq. 20). This implies no correlation between $\dot{S}_c$ and $\dot{S}_x$. In this case, the likelihood function $f_d(\hat{\boldsymbol{d}}|\boldsymbol{b}, N_s)$, where

$$\hat{\boldsymbol{d}} = (\hat{D}_{cc}, \hat{R}_{cx}, \hat{J}_{cx}, \hat{D}_{xx})^{\mathrm{T}}, \tag{28}$$

can be written as a multiplication of likelihood functions of individual components:

$$f_d(\hat{\boldsymbol{d}}|\boldsymbol{b}, N_s) = f(\hat{D}_{cc}|\boldsymbol{b}, N_s) f(\hat{R}_{cx}|\boldsymbol{b}, N_s) f(\hat{J}_{cx}|\boldsymbol{b}, N_s) f(\hat{D}_{xx}|\boldsymbol{b}, N_s). \tag{29}$$

PDFs of the individual components can be found as follows:

$$f\left(\hat{D}_{cc}|\boldsymbol{b}, N_s\right) = \frac{N_s}{\sigma_c^2} \chi^2_{2N_s}\left(\frac{N_s}{\sigma_c^2}\hat{D}_{cc}\right), \tag{30}$$

$$f\left(\hat{D}_{xx}|\boldsymbol{b}, N_s\right) = \frac{N_s}{\sigma_x^2} \chi^2_{2N_s}\left(\frac{N_s}{\sigma_x^2}\hat{D}_{xx}\right), \tag{31}$$

$$f\left(\hat{R}_{cx}|\boldsymbol{b}, N_s\right) = \frac{(2N_s)^a |\hat{R}_{cx}|^{-b}}{\sqrt{\pi} 2^{2N_s} (\sigma_c\sigma_x)^a \Gamma(N_s)} K_b\left(\frac{|2N_s\hat{R}_{cx}|}{\sigma_c\sigma_x}\right), \tag{32}$$

$$f\left(\hat{J}_{cx}|\boldsymbol{b}, N_s\right) = \frac{(2N_s)^a |\hat{J}_{cx}|^{-b}}{\sqrt{\pi} 2^{2N_s} (\sigma_c\sigma_x)^a \Gamma(N_s)} K_b\left(\frac{|2N_s\hat{J}_{cx}|}{\sigma_c\sigma_x}\right), \tag{33}$$

where $\chi^2_k$ is the chi-squared distribution with $k$ degrees of freedom,

$$a = (2N_s + 1)/2, \tag{34}$$
$$b = (1 - 2N_s)/2, \tag{35}$$

$\Gamma$ is the gamma function, and $K_\mu$ is the Bessel function of the second kind of order $\mu$. Recall, that $\sigma_c$ and $\sigma_x$ in Eqs. 30–33 are derived from the elements of $\boldsymbol{b}$ using Eqs. 21–22 and Eqs 24 and 25. Derivation and Monte Carlo evaluation of Eqs. 30–33 is given in Appendix B. Appendix B3 shows how to handle Eqs. 32 and 33 when $\hat{R}_{cx}$ and $\hat{J}_{cx}$ are close to 0.

## 3.3 Likelihood function in the h–v basis

Applying the rule of changing variables in a multivariate PDF (e.g. Walpole et al., 2012, Theorem 7.4) $f_b(\hat{\boldsymbol{b}}|\boldsymbol{b}, N_s)$ can be found from Eqs. 29 as follows:

$$f_b(\hat{\boldsymbol{b}}|\boldsymbol{b}, N_s) = f_d(\hat{\boldsymbol{d}}|\boldsymbol{b}, N_s). \tag{36}$$

As shown in Appendix B5, the determinant of the Jacobian of the transformation from $\hat{\boldsymbol{b}}$ to $\hat{\boldsymbol{d}}$ is equal to 1.

Likelihood $f_c(\hat{\boldsymbol{c}}|\boldsymbol{b}, N_s)$ of a vector

$$\hat{\boldsymbol{c}} = (\hat{B}_{hh}, \hat{Z}_{DR}, \hat{\rho}_{HV}, \hat{\Phi}_{DP}) \tag{37}$$





can be found by multiplying $f_b(\hat{\boldsymbol{b}}|\boldsymbol{b}, N_s)$ by $|\mathbf{J}_{cb}|$ with

$$\mathbf{J}_{cb} = -B_{hh}^3 Z_{DR}^{-3} \rho_{HV} \tag{38}$$

being the Jacobian of the transformation from $\hat{\boldsymbol{c}}$ to $\hat{\boldsymbol{b}}$ (see Appendix B6):

$$f_c(\hat{\boldsymbol{c}}|\boldsymbol{b}, N_s) = B_{hh}^3 Z_{DR}^{-3} \rho_{HV} f_b(\hat{\boldsymbol{b}}|\boldsymbol{b}, N_s). \tag{39}$$

Equations 36, and 39 can be used for the maximum likelihood optimization and Bayesian inference methods. Ready-to-use MATLAB implementations of these equations are provided in the supplement.

## 4 Error covariance matrices

A number of problems such as optimal estimation, data assimilation, and sensitivity analysis require the covariance matrix of the measurement errors. Unfortunately, an analytical integration of Eqs. 29, 36, and 39 required for the statistical moment calculation is challenging. In this section, however, known relations for calculation of variances and covariances after a linear transformation are used.

### 4.1 Error covariance matrix of $b$

The covariance matrix $\hat{\mathbf{B}}$ estimated from measurements is related to the matrix $\hat{\mathbf{D}}$ as follows:

$$\hat{\mathbf{B}} = \mathbf{Q}\hat{\mathbf{D}}\mathbf{Q}^\dagger. \tag{40}$$

Therefore, the elements of the vector $\hat{\boldsymbol{b}}$ can be found as linear combinations of the elements of the vector $\hat{\boldsymbol{d}}$:

$$\hat{B}_{hh} = q_{11}^2 \hat{D}_{cc} + |\dot{q}_{12}|^2 \hat{D}_{xx} + 2q_{11}\left(R_{12}\hat{R}_{cx} + J_{12}\hat{J}_{cx}\right), \tag{41}$$

$$\hat{R}_{hv} = q_{11}R_{12}\left(\hat{D}_{xx} - \hat{D}_{cc}\right) + \left(q_{11}^2 - R_{12}^2 + J_{12}^2\right)\hat{R}_{cx} - 2R_{12}J_{12}\hat{J}_{cx} \tag{42}$$

$$\hat{J}_{hv} = q_{11}J_{12}\left(\hat{D}_{xx} - \hat{D}_{cc}\right) + \left(q_{11}^2 + R_{12}^2 - J_{12}^2\right)\hat{J}_{cx} - 2R_{12}J_{12}\hat{R}_{cx} \tag{43}$$

$$\hat{D}_{vv} = |\dot{q}_{12}|^2 \hat{D}_{cc} + q_{11}^2\hat{D}_{xx} - 2q_{11}\left(R_{12}\hat{R}_{cx} + J_{12}\hat{J}_{cx}\right), \tag{44}$$

or in matrix form:

$$\hat{\boldsymbol{b}} = \begin{pmatrix} q_{11}^2 & 2q_{11}R_{12} & 2q_{11}J_{12} & |\dot{q}_{12}|^2 \\ -q_{11}R_{12} & q_{11}^2 - R_{12}^2 + J_{12}^2 & -2R_{12}J_{12} & q_{11}R_{12} \\ -q_{11}J_{12} & -2R_{12}J_{12} & q_{11}^2 + R_{12}^2 - J_{12}^2 & q_{11}J_{12} \\ |\dot{q}_{12}|^2 & -2q_{11}R_{12} & -2q_{11}J_{12} & q_{11}^2 \end{pmatrix} \hat{\boldsymbol{d}} = \mathbf{M}\hat{\boldsymbol{d}}. \tag{45}$$

In this case, as shown in Wilks D.S. (chapter 10.4.3), the error covariance matrix $\Sigma_b$ of $\hat{\boldsymbol{b}}$ can be calculated from the error 245 covariance matrix $\Sigma_d$ of $\hat{\boldsymbol{d}}$:

$$\Sigma_b = \mathbf{M}\Sigma_d\mathbf{M}^T, \tag{46}$$





where

$$
\Sigma_d = \begin{pmatrix}
4\sigma_c^4/N_s & 0 & 0 & 0 \\
0 & \sigma_c^2\sigma_x^2/N_s & 0 & 0 \\
0 & 0 & \sigma_c^2\sigma_x^2/N_s & 0 \\
0 & 0 & 0 & 4\sigma_x^4/N_s
\end{pmatrix}
\tag{47}
$$

The off-diagonal terms of $\Sigma_d$ are set to 0 taking into account that the elements of $\hat{d}$ are not correlated. The derivation of
250  diagonal terms – variances of elements of $\hat{d}$ – is given in Appendix C. A ready-to-use MATLAB implementation of Eq. 46 is
provided in the supplement.

## 4.2 Error covariance matrix of $c$

As it was shown in Sec. 4.1, the error covariance matrix $\Sigma_b$ can be used to characterize uncertainties of spectral radar ob-
servations. In this study, however, the error covariance of the vector $\hat{c}$ is also obtained. It will be further demonstrated that a
255  representation of measurement uncertainties for $\hat{c}$ is deficient.

Recall that the calculation of $\hat{c}$ includes highly nonlinear functions. Therefore, the error covariance matrix $\Sigma_c$ of the vector
$\hat{c}$ is estimated using the first-order Taylor approximation. Bringi and Chandrasekar (2001) used a similar approach to calculate
variances of polarimetric variables.

$$
\Sigma_c = \mathbf{S}\Sigma_b\mathbf{S}^T,
\tag{48}
$$

260  where $\mathbf{S}$ is the sensitivity matrix:

$$
\mathbf{S} = \begin{pmatrix}
\dfrac{\partial B_{hh}}{\partial B_{hh}} & \dfrac{\partial B_{hh}}{\partial R_{hv}} & \dfrac{\partial B_{hh}}{\partial J_{hv}} & \dfrac{\partial B_{hh}}{\partial B_{vv}} \\
\dfrac{\partial Z_{DR}}{\partial B_{hh}} & \dfrac{\partial Z_{DR}}{\partial R_{hv}} & \dfrac{\partial Z_{DR}}{\partial J_{hv}} & \dfrac{\partial Z_{DR}}{\partial B_{vv}} \\
\dfrac{\partial \rho_{HV}}{\partial B_{hh}} & \dfrac{\partial \rho_{HV}}{\partial R_{hv}} & \dfrac{\partial \rho_{HV}}{\partial J_{hv}} & \dfrac{\partial \rho_{HV}}{\partial B_{vv}} \\
\dfrac{\partial \Phi_{DP}}{\partial B_{hh}} & \dfrac{\partial \Phi_{DP}}{\partial R_{hv}} & \dfrac{\partial \Phi_{DP}}{\partial J_{hv}} & \dfrac{\partial \Phi_{DP}}{\partial B_{vv}}
\end{pmatrix}
\tag{49}
$$

Substituting Eqs. 10 – 12 into Eq. 49

$$
\mathbf{S} = \begin{pmatrix}
1 & 0 & 0 & 0 \\
B_{vv}^{-1} & 0 & 0 & -B_{hh}B_{vv}^{-2} \\
-0.5|\dot{B}_{hv}|B_{vv}^{-0.5}B_{hh}^{-1.5} & R_{hv}|\dot{B}_{hv}|^{-1}(B_{hh}B_{vv})^{-0.5} & J_{hv}|\dot{B}_{hv}|^{-1}(B_{hh}B_{vv})^{-0.5} & -0.5|\dot{B}_{hv}|B_{hh}^{-0.5}B_{vv}^{-1.5} \\
0 & -J_{hv}|\dot{B}_{hv}|^{-2} & R_{hv}|\dot{B}_{hv}|^{-2} & 0
\end{pmatrix}.
\tag{50}
$$

A ready-to-use MATLAB implementation of Eq. 48 is provided in the supplement.





**Table 1.** Operational setting of the used W-band radar

| Parameter | Chirp type 1 | Chirp type 2 | Chirp type 3 |
|---|---|---|---|
| Covered distance [km] | 0.1–1.2 | 1.2–4.9 | 4.9–15 |
| Range resolution [m] | 29.8 | 29.8 | 55 |
| Number of chirps in a sequence | 7168 | 7168 | 9216 |
| Chirp repetition frequency [kHz] | 9.2 | 7.5 | 5 |

## 5 Consistency checks on radar observations

In order to check consistency of Eqs. 36, 39, 46 and 48 with radar measurements, I/Q data collected with a W-band cloud radar with the hybrid polarimetric mode were used (Myagkov and Unal, 2021). The radar is a part of a dual-frequency system owned and operated by the Technical University of Delft in Cabauw, the Netherlands. Technical specifications of the radar can be found in Myagkov et al. (2020). The radar uses frequency modulated continuous signals. Küchler et al. (2017) explain the operation principle and shows that the radar profiles the atmosphere using several chirp types. Each chirp type is dedicated to a certain distance range. During measurements chirp types are switched consequently. For each chirp type a number of chirps (chirp sequence hereafter) is processed continuously. Operational settings used during I/Q measurements are listed in Table 1.

Measurements were made during a rain event on 21 June 2021 at 7:44 UTC. I/Q measurements provide high data rate of about 900 MB min$^{-1}$. Therefore, about 3 min of I/Q measurements were collected for the analysis. Since different chirp types have different properties, in the following only I/Q data collected with the first chirp type are used. Taking into account that the first chirp type has 37 range bins, in total $2.2 \times 10^3$ chirp sequences ($15.9 \times 10^6$ chirps) are available in each polarimetric channel.

### 5.1 Processing

All I/Q measurements within a chirp sequence in every polarimetric channel are split into 224 continuous blocks. Each block contains 32 I/Q pairs. The FFT with the Blackman weighting window is applied to each block to get complex Doppler spectra. Then the 224 blocks are split into 28 sub-blocks with 8 spectra in each sub-block. Within each sub-block elements of the vector $\hat{\boldsymbol{b}}$ are calculated according to Eqs. 15–18 with $N_s = 8$ for every spectral line. For each $\hat{\boldsymbol{b}}$ the vector $\hat{\boldsymbol{c}}$ is obtained. Note, that for this Eqs. 10–12 were applied to elements of $\hat{\boldsymbol{b}}$ instead of $\boldsymbol{b}$. Using vectors $\hat{\boldsymbol{b}}$ and $\hat{\boldsymbol{c}}$ within a sequence the error covariance matrices $\hat{\Sigma}_b$ and $\hat{\Sigma}_c$ are calculated numerically. The overhat here indicate that the error covariance matrices are estimated from measurements.

The calculation of the likelihood functions using Eqs. 36 and 39 require $\boldsymbol{b}$. The approximation of covariance matrices using Eqs. 46 and 48 requires the matrix **B**. In order to estimate $\boldsymbol{b}$ and **B**, elements of the vector $\hat{\boldsymbol{b}}$ are averaged over 28 sub-blocks





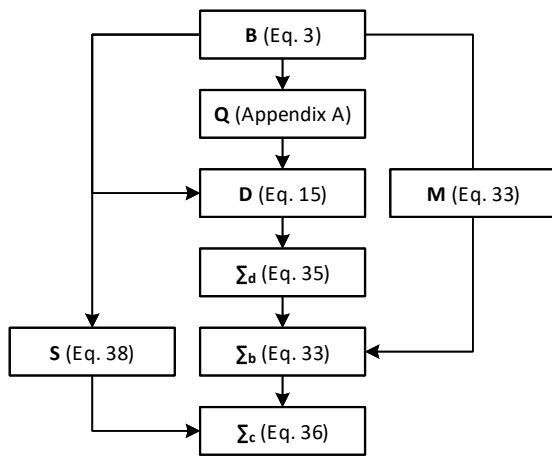

**Figure 1.** Schematic illustration of the error covariance matrix calculation.

available within a single chirp sequence. These averaged values are assumed to be elements of the vector $b$ from which the matrix $\mathbf{B}$ is obtained. Using $\mathbf{B}$ and $N_s = 8$, $\Sigma_b$ and $\Sigma_c$ are calculated for each chirp sequence as shown in Fig. 1.

## 5.2 Filtering

The random error analysis provided in this study is only applicable to volume-distributed scattering and noise. As discussed in Sec. 2, in this case $\hat{R}_h$ is not correlated with $\hat{J}_h$ and $\hat{R}_v$ is not correlated with $\hat{J}_v$. However, radar observations in general contain scattering from atmospheric plankton, ground clutter, and coherent receiver noise, which do not fulfil the assumption. In order to filter out spectral lines with correlated real and imaginary parts, a simple filtering rule was applied. It is known, that for a signal with uncorrelated in-phase and quadrature components, its mean power and power standard deviation are related to each other (Eq. 5.193 in Bringi and Chandrasekar, 2001). Figure 2 shows distributions of the mean power over the power standard deviation calculated in the horizontal and vertical polarization channels shown by blue and yellow lines, respectively. It can be seen that the mode of the distributions is close to the theoretical value of $\sqrt{N_s} = 2.8$. The distributions, however, have a considerable tail on the left side. These small values of the ratio are expected for correlated in-phase and quadrature components. Thus, a threshold in the ratio of the mean power over the standard deviation of power can be used to filter out unwanted spectral lines. In order to specify the threshold, the Monte Carlo approach was used. $15.9 \times 10^6$ random complex values with normal distribution, zero mean, and the standard deviation of 1 were generated. The same processing as for measured I/Q data was applied to the generated complex values. The distribution of the ratio of the mean power over the power standard deviation for the generated data (denoted as expected distribution) is shown in Fig. 2 by the red line. The expected distribution has much smaller tail on the left side relative to the ones of the measured distributions. The threshold of 2.3 used for filtering is chosen as the 5th percentile of the expected distribution. Vectors $\hat{b}$ and $\hat{c}$ are excluded from the analysis





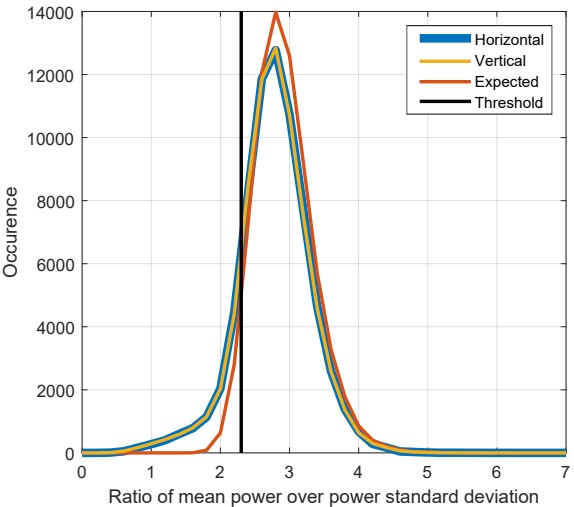

**Figure 2.** Distributions of the ratio of mean power over the power standard deviation for the horizontal (blue line) and vertical (yellow line) channels. The expected distribution is shown with the red line. The black vertical line indicates the threshold corresponding to the 5th percentile of the distribution for the randomly generated complex numbers.

if for the corresponding spectral component within a chirp sequence the ratio of the mean power over the power standard deviation is below the threshold in at least one of the polarimetric channels. The amount of excluded data is about 18 %.

### 5.3 Evaluation of $f_b(\hat{b}|b, N_s)$ and $f_b(\hat{c}|b, N_s)$

Recall, that $b$ is estimated from measurements by averaging all available sub-blocks within a chirp sequence. $b$, however, can also be estimated by maximization of the likelihood functions given in Eqs. 36 and 39. In this case, an optimization algorithm needs to be employed to find a set of elements of $b$ corresponding to the global maximum in either Eq. 36 or Eq. 39. This study uses a derivative-free optimization method available by default in MATLAB (Lagarias et al., 1998). Since the optimization method minimizes a function, the likelihood functions were not used directly. Instead, the following cost functions were used

for the minimization:

$$C_b = -\sum_{l=1}^{28} \log_{10}(f_b(\hat{b}|b, N_s)), \tag{51}$$

$$C_c = -\sum_{l=1}^{28} \log_{10}(f_c(\hat{c}|b, N_s)). \tag{52}$$

Here the index $l$ runs over 28 sub-blocks within a chirp sequence. Equations 51 and 52 take into account that the consecutive $\hat{b}$ are not correlated. In this case the total likelihood of 28 vectors $\hat{b}$ is a product of likelihood of each individual $\hat{b}$. In order to

avoid an overflow of double numbers, the logarithm was used. In this case the logarithm of the product is replaced by the sum of





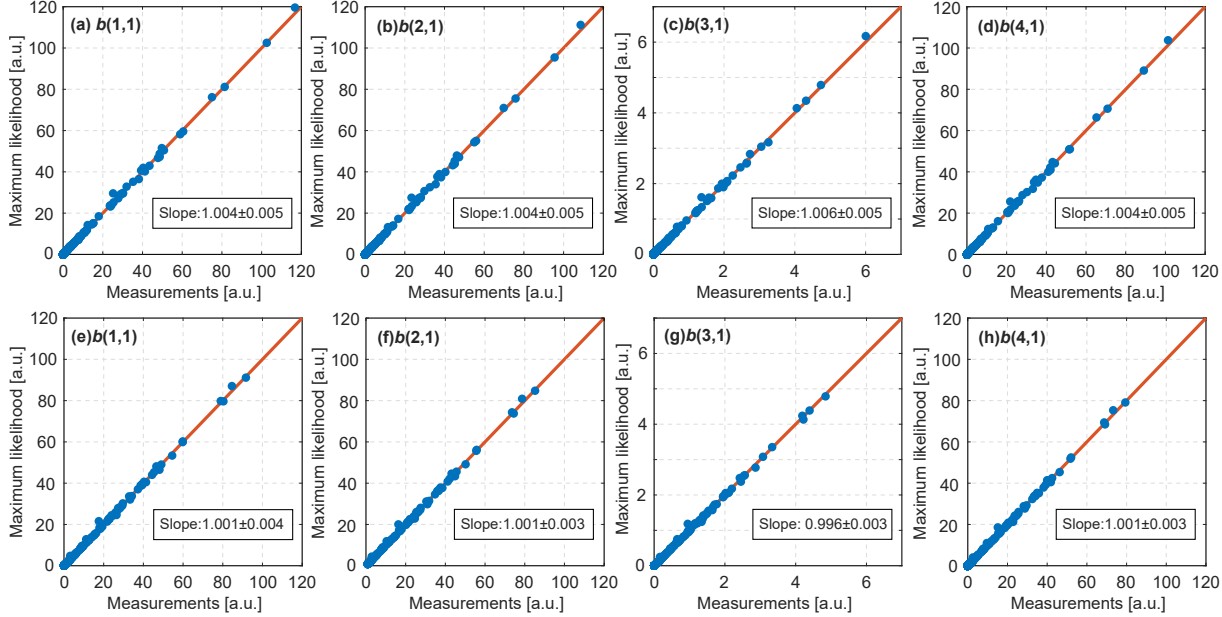

**Figure 3.** Comparison of elements of $\boldsymbol{b}$ estimated by the averaging over 28 sub-blocks (x-axis) with those estimated by the maximum likelihood approach (y-axis). $f_b(\hat{\boldsymbol{b}}|\boldsymbol{b}, N_s)$ was used for panels (a)–(d). $f_c(\hat{\boldsymbol{c}}|\boldsymbol{b}, N_s)$ was used for panels (e)–(h). Each panel contains 1000 points described in text. Linear regressions are shown by red solid lines. Each panel has a text box with the slope of the corresponding linear regression. Uncertainties of the slopes were estimated using the bootstrapping. Note, that units are not critical for the evaluation of the correctness of the derived likelihood functions. Therefore, arbitrary units (a.u) are used.

logarithms. The logarithm is monotonically increasing function and, therefore, it does not change the position of the maximum of the likelihood function. Finally, the minus sign was introduced to have a smaller value of a cost function corresponding to a higher value of the likelihood. For the evaluation, 1000 chirp sequences were chosen randomly for the maximum likelihood estimation using $f_b(\hat{\boldsymbol{b}}|\boldsymbol{b}, N_s)$. In each chirp sequence a single spectral line was randomly chosen for the analysis. Thus, there
are 28 vectors $\hat{\boldsymbol{b}}$ available in each of the 1000 chirp sequences. For each sequence, the optimization algorithm requires an initial guess of $\boldsymbol{b}$. In order to avoid local minima, 5 different initial guesses were used, which are a coefficient $P$ multiplied by the first $\hat{\boldsymbol{b}}$ in the analyzed chirp sequence. The values of $P$ were 0.5, 0.75, 1, 1.25, and 1.5. The solution giving the lowest cost function out of the 5 outcomes was chosen as the result. Similarly the maximum likelihood estimation using $f_c(\hat{\boldsymbol{c}}|\boldsymbol{b}, N_s)$ was done using independently chosen 1000 chirp sequences. Figure 3 shows a comparison of elements of $\boldsymbol{b}$ estimated by the averaging over
28 sub-blocks and those estimated by the maximum likelihood approach. All panels show a good agreement indicated by the close-to-unity slope of the linear regression. Both $f_b(\hat{\boldsymbol{b}}|\boldsymbol{b}, N_s)$ (results in the first row of Fig. 3) and $f_c(\hat{\boldsymbol{c}}|\boldsymbol{b}, N_s)$ (results in the second row of Fig. 3) show the same level of agreement and, therefore, can be used with no difference.





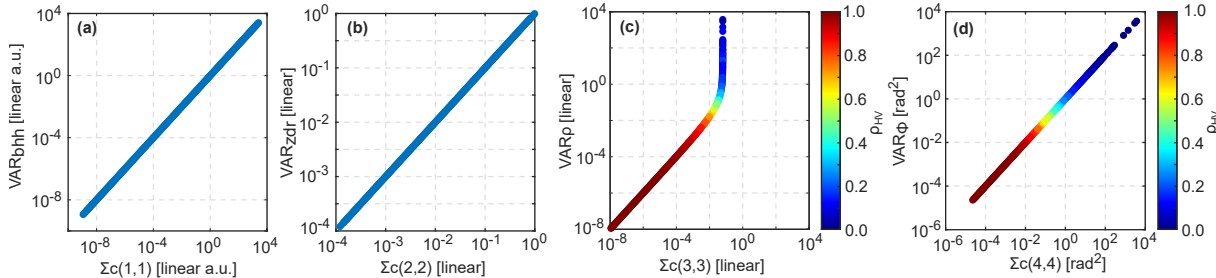

**Figure 4.** Comparison of variances of (a) $\hat{B}_{hh}$, (b) $\hat{Z}_{DR}$, (c) $\hat{\rho}_{HV}$, and (d) $\hat{\Phi}_{DP}$. Approximations developed in this study are on the x-axis. Approximations from Bringi and Chandrasekar (2001) are on the y-axis. $\rho_{HV}$ is color-coded in panels (c) and (d) to illustrate at which values of $\rho_{HV}$ approximations lead to erroneous values (see details in text). Note, that units are not critical for the evaluation of the derived equations. Therefore, arbitrary units (a.u) are used in the panel (a).

## 5.4 Evaluation of $\Sigma_c$

Diagonal elements of $\Sigma_c$ – variances of $\hat{B}_{hh}$, $\hat{Z}_{DR}$, $\hat{\rho}_{HV}$, and $\hat{\Phi}_{DP}$ – were checked against those calculated using Eqs. 6.139a,

6.141, 6.144, and 6.143 in Bringi and Chandrasekar (2001), respectively. Taking into account, that samples for a spectral line are not correlated, approximations for variances of $\hat{B}_{hh}$, $\hat{Z}_{DR}$, $\hat{\rho}_{HV}$, and $\hat{\Phi}_{DP}$ based on the equations in Bringi and Chandrasekar (2001) are:

$$\text{VAR}_{bhh} = \frac{B_{hh}^2}{N_s}, \tag{53}$$

$$\text{VAR}_{zdr} = \frac{2Z_{DR}^2(1-\rho_{HV}^2)}{N_s}, \tag{54}$$

$$\text{VAR}_\rho = \frac{(1-\rho_{HV}^2)^2}{2N_s\rho_{HV}^2}, \tag{55}$$

$$\text{VAR}_\Phi = \frac{(1-\rho_{HV}^2)}{2N_s\rho_{HV}^2}, \tag{56}$$

$$\tag{57}$$

respectively.

Figure 4 shows that $\text{VAR}_{bhh}$, $\text{VAR}_{zdr}$, and $\text{VAR}_\Phi$ match exactly $\Sigma_c(1,1)$, $\Sigma_c(2,2)$, and $\Sigma_c(4,4)$, respectively. $\text{VAR}_\rho$, how-

ever, agrees with $\Sigma_c(3,3)$ only at values of $\rho_{HV} > 0.95$. Below this value $\text{VAR}_\rho$ overestimates the variance of $\hat{\rho}_{HV}$. At values of $\rho_{HV}$ close to 0, $\text{VAR}_\rho$ has unrealistically high values, which result from $\rho_{HV}$ in the denominator of Eq. 55.

Figure 4d also shows unrealistic values with both approximations of the $\hat{\Phi}_{DP}$ variance. Taking into account that $\hat{\Phi}_{DP}$ can take values within the range from 0 to $2\pi$ rad, the variance of $\hat{\Phi}_{DP}$ exceeding $10^3$ rad$^2$ is definitely erroneous. The high variance of $\hat{\Phi}_{DP}$ corresponds to values of $\rho_{HV} < 0.3$. This effect results from the first-order Taylor approximation of Eq. 12

which is a highly non-linear function.

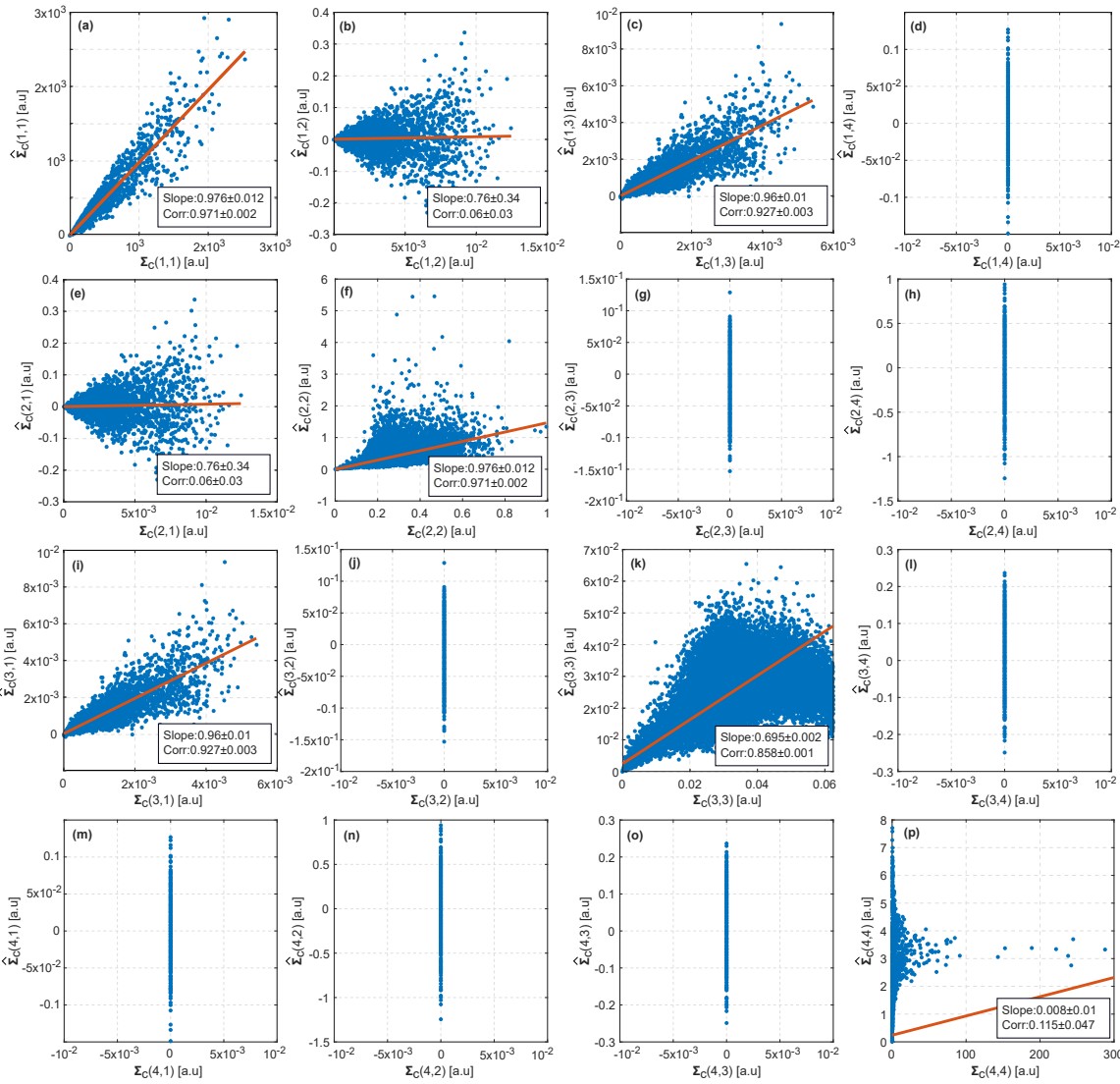

**Figure 5.** Comparison of $\hat{\Sigma}_c$ estimated from the radar measurements with $\Sigma_c$ obtained from Eq. 48. Elements of $\Sigma_c$ are given on the x-axes. Elements of $\hat{\Sigma}_c$ are given on the y-axes. The first and the second numbers in brackets indicate the row and the column of the corresponding matrix, respectively. Linear regressions are shown by red lines. Slopes of the linear regressions and Pearson correlations are given in boxes in each panel. Uncertainties in the slope and the correlation are represented by $\pm$ one standard deviation of the corresponding parameter. The standard deviations are obtained using the bootstrapping. Panels without linear regressions show elements for which Eq. 48 gives only near-zero values. Note, that units are not critical for the evaluation of the derived equations. Therefore, arbitrary units (a.u) are used. Also note that only values on the x and y axes on an individual panel should be compared. Value on different panels should not be compared.

A comparison of the error covariance matrices $\hat{\Sigma}_c$ with the calculated one $\Sigma_c$ is shown in Fig. 5. Panels (f), (k), and (p) indicate considerable differences caused by the first-order Taylor approximation in variances of $\hat{Z}_{DR}$, $\hat{\rho}_{HV}$, and $\hat{\Phi}_{DP}$, respec-





tively. The results also reveal that the first-order Taylor approximation cannot adequately represent most of the non-diagonal components of the error covariance matrix.

## 5.5 Evaluation of $\Sigma_b$

Figure 6 shows a comparison of elements of error covariance matrices $\hat{\Sigma}_b$ estimated from the radar measurements with those calculated using Eq. 46. Estimated and calculated elements are in a good agreement. Linear regressions shown in the panels by red lines have slopes close to 1. Pearson correlations between estimated and calculated elements exceed 0.96. These results indicate an agreement of the theoretical calculation with measurements and, thus, confirm correctness of Eq. 46. It is, thus, concluded that any application of spectral polarimetric measurements which require the estimate of the error covariance matrix (e.g. variational retrievals, data assimilation, and sensitivity analysis) should be performed in the space of observations $\hat{b}$ rather than $\hat{c}$.

## 6 Summary

Spectral and polarimetric cloud radar observations have a great potential in the cloud science (Kollias et al., 2020). Decades of such measurements have been already collected by e.g. the ARM (Atmospheric Radiation Measurement) and CLOUDNET communities. An advanced application of these vast datasets requires an accurate characterization of measurement uncertainties. Systematic errors in moment radar data and polarimetric variables have been discussed in many studies. Random measurement errors, in contrast, are rarely considered in literature. The are three main problems in existing random-error-characterization methods, namely (1) a lack of joint PDF for averaged polarimetric measurements, (2) neglection of non-diagonal components of the error covariance matrix, and (3) inaccuracy of the first-order approximation in variances of polarimetric variables. This study, thus, aims to provide solutions for these three problems.

Equations provided in Sec. 3 give an exact mathematical solution for the joint PDFs of spectral polarimetric observations. The PDFs are given for two equivalent representations of the measurements: (1) $b = (B_{hh}, R_{hv}, J_{hv}, B_{vv})^{\mathrm{T}}$, and (2) $c = (B_{hh}, Z_{DR}, \rho_{HV}, \Phi_{DP})^{\mathrm{T}}$. The obtained equations take into account non-coherent averaging of spectra, which is applied by a majority of cloud radars to improve the sensitivity. Maximum likelihood estimators of $b$ based on Eqs. 36 and 39 were compared with the estimator based on longer averaging. The comparison was based on dual-polarimetric cloud radar observations. The comparison showed a good agreement. Both PDFs can be equivalently used for methods based on the maximum likelihood and Bayesian inference.

Section 4 is focused on the error covarince matrix required for a number of applications such as data assimilation, sensitivity analysis, and variational retrievals. The error covariance matrices $\Sigma_b$ and $\Sigma_c$ for $b$ and $c$, respectively, are obtained using the characteristic functions of the PDFs described in Sec. 3. Since the calculation of the $c$ includes highly non-linear functions, $\Sigma_c$ was derived using the first-order Taylor approximation. The same approach was used by Bringi and Chandrasekar (2001) to get equations for variances of polarimetric observations.







**Figure 6.** Comparison of $\hat{\Sigma}_b$ estimated from the radar measurements with $\Sigma_b$ obtained from Eq. 46. Elements of $\Sigma_b$ are given on the x-axes. Elements of $\hat{\Sigma}_b$ are given on the y-axes. The first and the second numbers in brackets indicate the row and the column of the corresponding matrix, respectively. Linear regressions are shown by red lines. Slopes of the linear regressions and Pearson correlations are given in boxes in each panel. Uncertainties in the slope and the correlation are represented by $\pm$ one standard deviation of the corresponding parameter. The standard deviations are obtained using the bootstrapping. Note, that units are not critical for the evaluation of the derived equations. Therefore, arbitrary units (a.u) are used.





The error covariance matrices were evaluated using I/Q observations from a polarimetric W-band radar. It is illustrated
that elements of $\Sigma_c$ have considerable differences from those estimated from the measurements. The differences are related
to the first-order Taylor approximation which does not take into account non-linearities. In contrast, $\Sigma_b$ agrees well with the
observations. The correlation between calculated elements of $\Sigma_b$ with those estimated from the observations exceeds 0.965.

Thus, based on the results found within this study, it is recommended to use the vector $b$ to represent polarimetric cloud
radar observations for applications requiring the error covariance matrix. This representation has a better characterization of
random errors in comparison with widely used representation $c$.

*Code and data availability.* I/Q data used in this study are available on Zenodo (Myagkov and Unal, 2021, https://doi.org/10.5281/zenodo.5126813).
MATLAB code used to process I/Q data is provided in the supplement to this paper. Ready-to-use MATLAB implementations for Eqs. 36,
39, 46, and 48 are given in the supplement.

## Appendix A: Diagonalization matrix Q

The operator $\mathbf{Q}$, which is used to diagonalize the covariance matrix $\mathbf{B}$ in Eq. 20, is calculated as follows (Kanareykin et al.,
1968, chapter 2.5):

$$\mathbf{Q} = \begin{pmatrix} q_{11} & \dot{q}_{12} \\ -\dot{q}_{12}^* & q_{11} \end{pmatrix}, \tag{A1}$$

where

$$q_{11} = \left(1 + \left|\dot{d}\right|^2\right)^{-0.5}, \tag{A2}$$

$$\dot{q}_{12} = -\dot{d}^* q_{11}, \tag{A3}$$

$$d = \frac{\dot{B}_{hv}^*}{0.5\left[\mathrm{Tr}\mathbf{B} + \sqrt{\mathrm{Tr}\mathbf{B}^2 - 4\det(\mathbf{B})}\right] - B_{vv}}. \tag{A4}$$

In Eq. A4 Tr is the matrix trace.

## Appendix B: Derivation of likelihood functions

### B1 Change of variables in a PDF

Consider a vector $a$ with $n$ random variables $a_{1...n}$. Assume the joint PDF $f_a(a)$ of the variables is known. The joint PDF
$f_y(y)$ of a vector

$$y = G(a) \tag{B1}$$





can be found changing the variables in $f_a(\boldsymbol{a})$:

$$f_y(\boldsymbol{y}) = |\mathbf{J}| f_a \left[ G^{-1}(\boldsymbol{y}) \right], \tag{B2}$$

where $G^{-1}$ is reverse transformation from $\boldsymbol{y}$ to $\boldsymbol{a}$ and $\mathbf{J}$ is the determinant of the Jacobian of the transformation $\boldsymbol{a} = G^{-1}(\boldsymbol{y})$.

**B2   Likelihood functions for $D_{cc}$ and $D_{xx}$**

It is known, that the PDF of $z_s$ being a sum of squares of independent standard normal samples (i.e. distributed normally with 0 mean and standard deviation of 1) is the chi-squared distribution $\chi_k^2(z_s)$, where the degree of freedom $k$ shows how many samples have been summed. Taking into account that:

$$\hat{D}_{cc} = N_s^{-1}\sigma_c^2 \left\{ \sigma_c^{-2} \sum_{l=1}^{N_s} \mathrm{Re} \left( \dot{S}_c \right)_l^2 + \sigma_c^{-2} \sum_{l=1}^{N_s} \mathrm{Im} \left( \dot{S}_c \right)_l^2 \right\}, \tag{B3}$$

where the first and the second summed terms in the curly brackets are sums of squares of independent standard normal samples, the likelihood function $f\left( \hat{D}_{cc} | \sigma_c, N_s \right)$ can be found by changing the variable $z_s$ to $N_s \sigma_c^{-2} \hat{D}_c$:

$$f\left( \hat{D}_{cc} | \sigma_c, N_s \right) = \frac{N_s}{\sigma_c^2} \chi_{2N_s}^2 \left( \frac{N_s}{\sigma_c^2} \hat{D}_{cc} \right). \tag{B4}$$

The factor of 2 in the degree of freedom is because there are $2N_s$ summed components in the curly brackets in Eq. B3. The

equation for $\hat{D}_{xx}$ is derived in a similar manner as for $\hat{D}_{cc}$ resulting in:

$$f\left( \hat{D}_{xx} | \sigma_x, N_s \right) = \frac{N_s}{\sigma_x^2} \chi_{2N_s}^2 \left( \frac{N_s}{\sigma_x^2} \hat{D}_{xx} \right). \tag{B5}$$

**B3   Likelihood functions for $R_{cx}$ and $J_{cx}$**

Nadarajah and Pogány (2016) provide a solution for the PDF of an averaged multiplication $z_m$ of two standard normal variables. For two uncorrelated variables the PDF is defined as follows:

$$f_z(z_m) = \frac{n^{(n+1)/2} 2^{(1-n)/2} |z_m|^{(n-1)/2}}{\sqrt{\pi} \Gamma(n/2)} K_{(1-n)/2}\left( n|z_m| \right), \tag{B6}$$

where $n$ is the number of averaged multiplications, $\Gamma$ is the gamma function, and $K_\mu$ is the Bessel function of the second kind of order $\mu$.

$\hat{R}_{cx}$ is calculated as follows:

$$\hat{R}_{cx} = 2\sigma_c \sigma_x \left\{ \frac{1}{2N_s \sigma_c \sigma_x} \left[ \sum_{l=1}^{N_s} \mathrm{Re}\left( \dot{S}_c \right) \mathrm{Re}\left( \dot{S}_x \right) + \sum_{l=1}^{N_s} \mathrm{Im}\left( \dot{S}_c \right) \mathrm{Im}\left( \dot{S}_x \right) \right] \right\}, \tag{B7}$$

where the term in the curly brackets is an average over $2N_s$ multiplications of independent standard normal samples. In this case, the likelihood function $f\left( \hat{R}_{cx} | \sigma_c, \sigma_x, N_s \right)$ can be found by changing $z_m$ by $(2\sigma_c \sigma_x)^{-1} \hat{R}_{cx}$:

$$f\left( \hat{R}_{cx} | \sigma_c, \sigma_x, N_s \right) = \frac{(2N_s)^a |\hat{R}_{cx}|^{-b}}{\sqrt{\pi} 2^{2N_s} (\sigma_c \sigma_x)^a \Gamma(N_s)} K_b \left( \frac{N_s |\hat{R}_{cx}|}{\sigma_c \sigma_x} \right), \tag{B8}$$





where $a = (2N_s + 1)/2$, $b = (1 - 2N_s)/2$, $\Gamma$ is the gamma function, and $K_\mu$ is the Bessel function of the second kind of order $\mu$. When $\hat{R}_{cx} \to 0$, the modified Bessel function $K_b\left(N_s|\hat{R}_{cx}|(\sigma_c\sigma_x)^{-1}\right) \to \infty$. Therefore, for $\hat{R}_{cx}$ close to 0, the following

approximation based on Eqs. 9.6.6 and 9.6.8 from Abramowitz and Stegun (1972) should be used:

$$f\left(\hat{R}_{cx}|\sigma_c, \sigma_x, N_s\right) \approx \frac{N_s\Gamma(-b)}{2\sqrt{\pi}\sigma_c\sigma_x\Gamma(N_s)}. \tag{B9}$$

Formulas for $\hat{J}_{cx}$ are defined in a similar manner:

$$f\left(\hat{J}_{cx}|\sigma_c, \sigma_x, N_s\right) = \frac{(2N_s)^a|\hat{J}_{cx}|^{-b}}{\sqrt{\pi}2^{2N_s}(\sigma_c\sigma_x)^a\Gamma(N_s)}K_b\left(\frac{N_s|\hat{J}_{cx}|}{\sigma_c\sigma_x}\right), \tag{B10}$$

with the approximation for $\hat{J}_{cx}$ close to 0:

$$f\left(\hat{J}_{cx}|\sigma_c, \sigma_x, N_s\right) \approx \frac{N_s\Gamma(-b)}{2\sqrt{\pi}\sigma_c\sigma_x\Gamma(N_s)}. \tag{B11}$$

## B4 Monte Carlo evaluation of Eqs. B4, B5, B8, and B10

For the equation evaluation a simulated dataset was generated. In total 1000 sets of distributions were simulated using the Monte Carlo approach. A single set included distributions of $\hat{B}_{cc}$, $\hat{B}_{xx}$, $\hat{R}_{cx}$, and $\hat{J}_{cx}$. For a single set $10^5$ vectors $\hat{b}$ were generated. A single vector $\hat{b}$ resulted from $N_s$ randomly generated vectors $m$. For a single set of distributions a single covariance matrix $\mathbf{B}$

was taken. The elements of the covariance matrix $\mathbf{B}$ and $N_s$ were randomly generated according to the following rules (values have linear arbitrary units):

1. $B_{hh}$ is a sum of mean powers of signal $P_{sh}$ and noise $P_{nh}$.

2. $B_{vv}$ is a sum of mean powers of signal $P_{sv}$ and noise $P_{nv}$.

3. $P_{nh} = P_{nv} = 1$

4. $P_{sh}$ and $P_{sv}$ were randomly and independently generated using the uniform distribution from 1 to 5.

5. $\dot{B}_{hv}$ was calculated as $\rho_{HV}e^{i\Phi_{DP}}\sqrt{P_{sh}P_{sv}}$.

6. $\rho_{HV}$ was chosen randomly using the uniform distribution from 0 to 1.

7. $\Phi_{DP}$ was chosen randomly using the uniform distribution from 0 to $2\pi$.

8. $N_s$ was chosen as a random integer number in the range from 2 to 80.

From the covariance matrix $\mathbf{B}$ the true covariance matrix $\Sigma_m$ was obtained. $10^5 \times N_s$ vectors $m$ were generated according to the PDF given in Eq. 2. Then, $10^5$ elements of the $\hat{b}$ were calculated according to Eqs. 15–18. Elements of the vector $\hat{d}$ were derived from the vectors $\hat{b}$ using Eq. 27.

Using the $10^5$ vectors $\hat{d}$ individual histograms for each of the variables $\hat{B}_{cc}$, $\hat{B}_{xx}$, $\hat{R}_{cx}$, and $\hat{J}_{cx}$ are derived. A histogram has 10 bins covering the range from the minimum to maximum values of the corresponding variable. Widths of bins were adjusted





**Table B1.** Percentage of test-statistic values exceeding critical values for different significance levels. Percentages are given in %. Names of 4 columns on the right side of the table indicate for which distribution a percentage is given.

| Significance level | Critical value | $f\left(\hat{D}_{cc}\vert\sigma_c, N_s\right)$ | $f\left(\hat{R}_{cx}\vert\sigma_c, N_s\right)$ | $f\left(\hat{J}_{cx}\vert\sigma_c, N_s\right)$ | $f\left(\hat{D}_{xx}\vert\sigma_c, N_s\right)$ |
|---|---|---|---|---|---|
| 0.95 | 16.919 | 6.9 | 5.2 | 5.8 | 5.9 |
| 0.975 | 19.023 | 3.8 | 3.4 | 2.7 | 2.9 |
| 0.99 | 21.666 | 1.0 | 1.3 | 1.0 | 1.2 |

to have 10000 samples in each bin. For the same bins the expected number of samples is calculated using the corresponding PDF. Since integration of Eqs. B4, B5, B8, and B10 is challenging, the integration is done numerically. Then the Pearson's chi-squared test is applied. The same procedure is repeated for all 1000 sets of distributions. Thus, for each PDF (Eqs. B4, B5, B8, and B10) 1000 test-statistic values were obtained.

The Pearson's chi-squared test implies a comparison of the test-statistic values with a critical values for a given level of
significance. A test-statistic value exceeding the critical value would indicate that there is a chance (equal to the significance level) that the data significantly differs from the PDF. There is, however, a small chance that the conclusion that the data differs from the PDF is erroneous. Table B1 shows the percentage of the test-statistic values exceeding critical values. It can be seen that the amount of test-statistic values exceeding corresponding critical values is very close to the theoretical values, i.e. 5, 2.5, and 1 % at 0.95, 0.975, and 0.99 significance levels, respectively. This confirms the validity of the obtained PDFs.

**B5   Jacobian $\mathbf{J}_{bd}$ of the transformation from $\hat{b}$ to $\hat{d}$**

Using Eqs. 21–22 $\mathbf{J}_{bd}$ can be written as follows:

$$
\mathbf{J}_{bd} =
\begin{vmatrix}
\dfrac{\partial D_{cc}}{\partial B_{hh}} & \dfrac{\partial D_{cc}}{\partial R_{hv}} & \dfrac{\partial D_{cc}}{\partial J_{hv}} & \dfrac{\partial D_{cc}}{\partial B_{vv}} \\
\dfrac{\partial R_{cx}}{\partial B_{hh}} & \dfrac{\partial R_{cx}}{\partial R_{hv}} & \dfrac{\partial R_{cx}}{\partial J_{hv}} & \dfrac{\partial R_{cx}}{\partial B_{vv}} \\
\dfrac{\partial J_{cx}}{\partial B_{hh}} & \dfrac{\partial J_{cx}}{\partial R_{hv}} & \dfrac{\partial J_{cx}}{\partial J_{hv}} & \dfrac{\partial J_{cx}}{\partial B_{vv}} \\
\dfrac{\partial D_{xx}}{\partial B_{hh}} & \dfrac{\partial D_{xx}}{\partial R_{hv}} & \dfrac{\partial D_{xx}}{\partial J_{hv}} & \dfrac{\partial D_{xx}}{\partial B_{vv}}
\end{vmatrix} =
$$

$$
\begin{vmatrix}
q_{11}^2 & |\dot{q}_{12}|^2 & -2q_{11}R_{12} & -2q_{11}J_{12} \\
|\dot{q}_{12}|^2 & q_{11}^2 & 2q_{11}R_{12} & 2q_{11}J_{12} \\
q_{11}R_{12} & -q_{11}R_{12} & q_{11}^2 - R_{12}^2 + J_{12}^2 & -2R_{12}J_{12} \\
q_{11}J_{12} & -q_{11}J_{12} & -2R_{12}J_{12} & q_{11}^2 + R_{12}^2 - J_{12}^2
\end{vmatrix}
= (q_{11}^2 + |q_{12}|^2)^4. \quad \text{(B12)}
$$

Taking into account Eqs. A2 and A3, $\mathbf{J}_{bd} = 1$.





### B6  Jacobian $\mathbf{J}_{cb}$ of the transformation from $\hat{c}$ to $\hat{b}$

Using Eqs. 21–22 $\mathbf{J}_{cb}$ can be written as follows:

$$\mathbf{J}_{cb} = \begin{vmatrix} \dfrac{\partial B_{hh}}{\partial B_{hh}} & \dfrac{\partial B_{hh}}{\partial Z_{DR}} & \dfrac{\partial B_{hh}}{\partial \rho_{HV}} & \dfrac{\partial B_{hh}}{\partial \Phi_{DP}} \\[2mm] \dfrac{\partial R_{hv}}{\partial B_{hh}} & \dfrac{\partial R_{hv}}{\partial Z_{DR}} & \dfrac{\partial R_{hv}}{\partial \rho_{HV}} & \dfrac{\partial R_{hv}}{\partial \Phi_{DP}} \\[2mm] \dfrac{\partial J_{hv}}{\partial B_{hh}} & \dfrac{\partial J_{hv}}{\partial Z_{DR}} & \dfrac{\partial J_{hv}}{\partial \rho_{HV}} & \dfrac{\partial J_{hv}}{\partial \Phi_{DP}} \\[2mm] \dfrac{\partial B_{vv}}{\partial B_{hh}} & \dfrac{\partial B_{vv}}{\partial R_{hv}} & \dfrac{\partial B_{vv}}{\partial J_{hv}} & \dfrac{\partial B_{vv}}{\partial B_{vv}} \end{vmatrix} =$$

$$= \begin{vmatrix} 1 & 0 & 0 & 0 \\ \rho_{HV}\cos(\Phi_{DP})Z_{DR}^{-0.5} & -0.5B_{hh}\rho_{HV}\cos(\Phi_{DP})Z_{DR}^{-1.5} & B_{hh}\cos(\Phi_{DP})Z_{DR}^{-0.5} & -B_{hh}\rho_{HV}\sin(\Phi_{DP})Z_{DR}^{-0.5} \\ \rho_{HV}\sin(\Phi_{DP})Z_{DR}^{-0.5} & -0.5B_{hh}\rho_{HV}\sin(\Phi_{DP})Z_{DR}^{-1.5} & B_{hh}\sin(\Phi_{DP})Z_{DR}^{-0.5} & -B_{hh}\rho_{HV}\cos(\Phi_{DP})Z_{DR}^{-0.5} \\ Z_{DR}^{-1} & -B_{hh}Z_{DR}^{-2} & 0 & 0 \end{vmatrix} =$$

$$- B_{hh}^3 Z_{DR}^{-3}\rho_{HV} \quad \text{(B13)}$$

## Appendix C:  Variances of elements of the vector $\hat{d}$

To derive solutions for mean and variances of elements of $\hat{d}$, the distribution of the elements is represented by characteristic functions. A $\gamma$-th raw statistical moment $M_\gamma$ of a random variable with a characteristic function $\phi(t)$ can be found as follows:

$$M_\gamma = i^{-\gamma}\frac{d^\gamma\phi(t)}{dt^\gamma}\bigg|_{t=0}, \quad \text{(C1)}$$

The calculation of derivatives of the characteristic functions is in general easier to obtain than integration of the corresponding PDFs.

The characteristic function of the chi-squared distribution $\chi_k^2(z_s)$ is

$$\phi_s(t) = (1-2it)^{-k/2}. \quad \text{(C2)}$$

Therefore, the characteristic function for $\hat{D}_{cc}$ for a given $\sigma_c$ and $N_s$ can be written in the following way:

$$\phi_{cc}(t) = \left(1 - \frac{2i\sigma_c^2 t}{N_s}\right)^{-N_s} \quad \text{(C3)}$$

The mean value and variance of $\hat{D}_{cc}$ are calculated as follows:

$$\overline{\hat{D}_{cc}} = \frac{1}{i}\frac{d\phi_{cc}(t)}{dt}\bigg|_{t=0} = 2\sigma_c^2, \quad \text{(C4)}$$

$$\text{var}(\hat{D}_{cc}) = -\frac{d^2\phi_{cc}(t)}{dt^2}\bigg|_{t=0} - \overline{\hat{D}_{cc}}^2 = \frac{4\sigma_c^4}{N_s}. \quad \text{(C5)}$$





Similarly,

$$\overline{\hat{D}_{xx}} = \frac{1}{i} \left. \frac{d\phi_{xx}(t)}{dt} \right|_{t=0} = 2\sigma_x^2, \tag{C6}$$

$$\mathrm{var}(\hat{D}_{xx}) = \frac{4\sigma_x^4}{N_s}. \tag{C7}$$

Based on Nadarajah and Pogány (2016) the characteristic function corresponding to $f_z(z_m)$ is

$$\phi_z(t) = \left(1 + \frac{t^2}{n^2}\right)^{-n/2}. \tag{C8}$$

Therefore, the characteristic function for $\hat{R}_{cx}$ and $\hat{J}_{cx}$ for given $\sigma_c$, $\sigma_x$, and $N_s$ is as follows:

$$\phi_{cx}(t) = \left(1 + \frac{\sigma_c^2 \sigma_x^2 t^2}{N_s^2}\right)^{-N_s/2}. \tag{C9}$$

As expected for a multiplication of two uncorrelated variables, the mean values of $\hat{R}_{cx}$ and $\hat{J}_{cx}$

$$\overline{\hat{R}_{cx}} = \overline{\hat{J}_{cx}} = \frac{1}{i} \left. \frac{d\phi_{cx}(t)}{dt} \right|_{t=0} = 0. \tag{C10}$$

The variance of $\hat{R}_{cx}$ and $\hat{J}_{cx}$ can be found as follows:

$$\mathrm{var}(\hat{R}_{cx}) = \mathrm{var}(\hat{J}_{cx}) = - \left. \frac{d^2\phi_{cx}(t)}{dt^2} \right|_{t=0} = \frac{\sigma_c^2 \sigma_x^2}{2N_s}. \tag{C11}$$

**Appendix D:  Table of symbols**

*Author contributions.* AM derived equations for PDFs and error covariance matrices, made evaluation using the radar observations, and prepared the first draft of the manuscript. DO reviewed the draft and essentially improved the manuscript.

*Competing interests.* AM is an employee of Radiometer Physics GmbH. DO declares no competing interests.

*Acknowledgements.* This work was carried out as a collaboration within the IMPRINT (Understanding Ice Microphysical Processes by combining multi-frequency and spectral Radar polarImetry aNd super-parTicle modelling, Project Number 408011764) project, which is a part of the German Research Foundation (DFG) Priority Program SPP2115 PROM (Fusion of Radar Polarimetry and Numerical Atmospheric Modelling Towards an Improved Understanding of Cloud and Precipitation Processes). The authors acknowledge Ruisdael Observatory, the Netherlands and Christine Unal from TU Delft for granting an access to the W-band radar in Cabauw to collect I/Q data used in this study.

Work of DO is funded by the German Research Foundation (DFG) under the grant SCHE 2074/1-1 (SPP HALO).



**Table D1.** Main symbols used throughout the study. The overdot indicates a complex number. Indices $h$ and $v$ indicate the polarization of the receiver channel. The overhat indicates a measured quantity.

| Symbol | Description |
|---|---|
| $\Gamma$ | Gamma function |
| $\rho_{HV}$ and $\hat{\rho}_{HV}$ | Correlation coefficient for a spectral line |
| $\Phi_{DP}$ and $\hat{\Phi}_{DP}$ | Differential phase for a spectral line |
| $\sigma_h$ | Standard deviation of $\hat{R}_h$ and $\hat{J}_h$ |
| $\sigma_v$ | Standard deviation of $\hat{R}_v$ and $\hat{J}_v$ |
| $\sigma_c$ | Standard deviation of $\hat{R}_c$ and $\hat{J}_c$ |
| $\sigma_x$ | Standard deviation of $\hat{R}_x$ and $\hat{J}_x$ |
| $\Sigma_m$ | Error covariance matrix of $\hat{\boldsymbol{m}}$ |
| $\Sigma_d$ | Error covariance matrix of $\hat{\boldsymbol{d}}$ |
| $\Sigma_b$ and $\hat{\Sigma}_b$ | Error covariance matrix of $\hat{\boldsymbol{b}}$ |
| $\Sigma_c$ and $\hat{\Sigma}_c$ | Error covariance matrix of $\hat{\boldsymbol{c}}$ |
| $\phi(t)$ | characteristic function |
| $\phi_s(t)$ | characteristic function for $z_s$ |
| $\phi_{cc}(t)$ | characteristic function for $\hat{D}_{cc}$ |
| $\phi_{cx}(t)$ | characteristic function for $\hat{R}_{cx}$ and $\hat{J}_{cx}$ |
| $\phi_z(t)$ | characteristic function for $z_m$ |
| $\chi^2_k$ | chi-squared distribution with $k$ degrees of freedom |
| $*$ | Complex conjugation sign |
| $\dagger$ | Hermitian conjugate sign |
| $\boldsymbol{b}$ | the column vector elements of which are $B_{hh}$, $R_{hv}$, $J_{hv}$, and $B_{vv}$ |
| $\hat{\boldsymbol{b}}$ | the column vector elements of which are $\hat{B}_{hh}$, $\hat{R}_{hv}$, $\hat{J}_{hv}$, and $\hat{B}_{vv}$ |
| $\mathbf{B}$ and $\hat{\mathbf{B}}$ | $2 \times 2$ covariance matrix describing polarimetric measurements in a single spectral line in the $h - v$ basis |
| $B_{hh}$, $\dot{B}_{hv}$, and $B_{vv}$ | elements of the covariance matrix $\mathbf{B}$ |
| $\hat{B}_{hh}$ and $\hat{B}_{vv}$ | diagonal elements of the covariance matrix $\hat{\mathbf{B}}$ |
| $\boldsymbol{c}$ | the column vector elements of which are $B_{hh}$, $Z_{DR}$, $\rho_{HV}$, and $\Phi_{DP}$ |
| $\hat{\boldsymbol{c}}$ | the column vector elements of which are $\hat{B}_{hh}$, $\hat{Z}_{DR}$, $\hat{\rho}_{HV}$, and $\hat{\Phi}_{DP}$ |



**Table D2.** Continue of Table D1.

| Symbol | Description |
|---|---|
| $\hat{\boldsymbol{d}}$ | Column vector elements of which are $\hat{D}_{cc}$, $\hat{R}_{cx}$, $\hat{J}_{cx}$, and $\hat{D}_{xx}$ |
| $D_{cc}$ and $D_{xx}$ | Diagonal elements of the covariance matrix $\mathbf{D}$ |
| $\hat{D}_{cc}$, $\hat{D}_{vv}$, and $\hat{D}_{cx}$ | Elements of the covariance matrix $\hat{\mathbf{D}}$ |
| $\mathbf{D}$ and $\hat{\mathbf{D}}$ | $2 \times 2$ covariance matrix describing polarimetric measurements in a single spectral line in the $c - x$ basis |
| $\boldsymbol{e}$ | measurement column vector in the $h - v$ basis |
| $\boldsymbol{e}_D$ | measurement column vector in the $c - x$ basis |
| $f(\hat{D}_{cc}|\boldsymbol{b}, N_s)$ | PDF of $\hat{D}_{cc}$ for a given $\boldsymbol{b}$ and $N_s$ |
| $f(\hat{R}_{cx}|\boldsymbol{b}, N_s)$ | PDF of $\hat{R}_{cx}$ for a given $\boldsymbol{b}$ and $N_s$ |
| $f(\hat{J}_{cx}|\boldsymbol{b}, N_s)$ | PDF of $\hat{J}_{cx}$ for a given $\boldsymbol{b}$ and $N_s$ |
| $f(\hat{D}_{xx}|\boldsymbol{b}, N_s)$ | PDF of $\hat{D}_{xx}$ for a given $\boldsymbol{b}$ and $N_s$ |
| $f_m(\hat{\boldsymbol{m}}|\Sigma_m)$ | joint PDF of $\hat{\boldsymbol{m}}$ for a given $\Sigma_m$ |
| $f_d(\hat{\boldsymbol{d}}|\boldsymbol{b}, N_s)$ | joint PDF of $\hat{\boldsymbol{d}}$ for a given $\boldsymbol{b}$ and $N_s$ |
| $f_b(\hat{\boldsymbol{b}}|\boldsymbol{b}, N_s)$ | joint PDF of $\hat{\boldsymbol{b}}$ for a given $\boldsymbol{b}$ and $N_s$ |
| $f_c(\hat{\boldsymbol{c}}|\boldsymbol{b}, N_s)$ | joint PDF of $\hat{\boldsymbol{c}}$ for a given $\boldsymbol{b}$ and $N_s$ |
| $i$ | Imaginary unit |
| $I_{h,v}$ | Measured in-phase component measured by the radar receiver in a range bin |
| $J_{12}$ | Imaginary part of $\dot{q}_{12}$ |
| $\hat{J}_{cx}$ | Imaginary part of $\hat{D}_{cx}$ |
| $\hat{J}_h$ and $\hat{J}_v$ | Imaginary parts of $\dot{S}_h$ and $\dot{S}_v$, respectively |
| $J_{hv}$ | Imaginary part of $\dot{B}_{hv}$ |
| $\hat{J}_{hv}$ | Imaginary part of the covariance between $\dot{S}_h$ and $\dot{S}_v$ |
| $\mathbf{J}_{bd}$ | Jacobian of the transformation from $\hat{\boldsymbol{b}}$ to $\hat{\boldsymbol{d}}$ |
| $\mathbf{J}_{cb}$ | Jacobian of the transformation from $\hat{\boldsymbol{c}}$ to $\hat{\boldsymbol{b}}$ |
| $K_\mu$ | Bessel function of the second kind of order $\mu$ |
| $\hat{\boldsymbol{m}}$ | Measurement vector, which elements are real and imaginary parts of $\dot{S}_h$ and $\dot{S}_v$ |
| $M_\gamma$ | $\gamma$-th raw statistical moment of a random variable |
| $N_s$ | Number of spectra used for averaging |
| $N_{\text{fft}}$ | Number of pulses/chirps used to calculate the Doppler spectra |
| $Q_{h.v}$ | Measured quadrature component measured by the radar receiver in a range bin |





**Table D3.** Continue of Table D1.

| Symbol | Description |
| --- | --- |
| $\mathbf{Q}$ | Matrix used to diagonalize the matrix $\mathbf{B}$ |
| $q_{11}$, $\dot{q}_{12}$, and $q_{22}$ | Elements of the matrix $\mathbf{Q}$ |
| $q$ | correlation between $\hat{R}_h$ and $\hat{R}_v$ |
| $s$ | correlation between $\hat{R}_h$ and $\hat{J}_v$ |
| $R_{12}$ | real part of $\dot{q}_{12}$ |
| $\hat{R}_h$ and $\hat{R}_v$ | real parts of $\dot{S}_h$ and $\dot{S}_v$, respectively |
| $R_{hv}$ | real part of $\dot{B}_{hv}$ |
| $\hat{R}_{hv}$ | Real part of the covariance between $\dot{S}_h$ and $\dot{S}_v$ |
| $\hat{R}_{cx}$ | Real part of $\hat{D}_{cx}$ |
| $\dot{S}_{h,v}$ | Measured complex amplitude for a spectral line |
| $\mathbf{S}$ | the $4 \times 4$ sensitivity matrix |
| T | the transposition sign |
| t | argument of a characteristic function |
| $\text{VAR}_{bhh}$ | Variance of $\hat{B}_{hh}$ approximated from Bringi and Chandrasekar (2001) |
| $\text{VAR}_{zdr}$ | Variance of $\hat{Z}_{DR}$ approximated from Bringi and Chandrasekar (2001) |
| $\text{VAR}_\rho$ | Variance of $\hat{\rho}_{HV}$ approximated from Bringi and Chandrasekar (2001) |
| $\text{VAR}_\Phi$ | Variance of $\hat{\Phi}_{DP}$ approximated from Bringi and Chandrasekar (2001) |
| $Z_{DR}$ and $\hat{Z}_{DR}$ | Differential reflectivity for a spectral line |
| $z_s$ | a sum of squares of independent standard normal samples |
| $z_m$ | averaged multiplication of two standard normal variables |

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
