# Peer review of "Analytic characterization of random errors in spectral dual-polarized cloud radar observations"

_Atmospheric Measurement Techniques, 2021_

## Referee Comment (RC2)

This manuscript presents derivation of formulations for analytic treatment of error covariance matrix of spectral dual-polarization observations. This is an important study and of interest to the radar meteorology community. I would like to congratulate the authors on the heroic job of deriving theoretical formulations. I believe that it should be published in AMT after comments are addressed.

The main problem of this manuscript, it is not easy to read. Just to give you a few examples:
- In equation (9) you use Bhh, Zdr, rho_hv and Phi_dp. Why do you use Bhh and not Zhh, which would be more commonly used? Of course, Bhh is not Zhh, it is a spectral Zhh. But the same applies to Zdr and the rest of variables. It would be good if you would try to use more widely used notations.
- On line 89, page 3. You state "For each transmitted pulse (the term pulse is used throughout the study, although for radars with frequency modulated continuous wave signals a chirp would have been implied)". Then in Section 5 you return to using chirp and introduce chirp sequence. It took me some effort to adjust to that transition. I would suggest that you either use pulses or chirps.

The other concern is whether results of this study will ever be used. Because the formulations are rather complex, there is a good chance that they will never be adopted. Is there an approximation that can be used and that would work for most applications? If yes, could you make a recommendation. Of course, the other option could be a follow up study, demonstrating practical applications.

**Abstract:** "This study presents the first-ever complete characterization of random errors in dual-polarimetric spectral observations of meteorological targets by cloud radars."

While this statement is true, at least to my knowledge, it seems to me that the underlying assumptions are not very, if at all, different from what are used to describe normal dual-polarization observations. Could you please elaborate what are the main differences?
I would argue that the derived expressions presented in this study are representing a subset of conditions for which expressions in (Doviak and Zrnic) or (Chandrasekar and Bringi) are derived. One big simplification, which I believe is valid for spectra observations at least to some extent, used in this study is that averaging is performed using independent spectra, i.e. Ns represents number of independent averages. That means that one does not need to consider the correlation function, which is a must for time-domain observations. I think a more complete discussion on this should be included in the article. You are giving a short discussion on line 105. I would suggest that you compare and contrast the assumptions used in this study and more classical studies.

"One of the main conclusions of the study is that the convenient representation of spectral polarimetric measurements including differential reflectivity ZDR, correlation coefficient ρHV, and differential phase ΦDP is not suited for the proper characterization of the error covariance matrix."

I believe you need to explain this point more explicitly. In the summary the discussion is rather short:

"It is illustrated that elements of Σc have considerable differences from those estimated from the measurements. The differences are related to the first-order Taylor approximation which does not take into account non-linearities. In contrast, Σb agrees well with the observations. The correlation between calculated elements of Σb with those estimated from the observations exceeds 0.965."

I did not understand if the problem is due to the first-order Taylor approximation used or something else. In many applications an approximation would perfectly well. What would you recommend using to get an estimate of the uncertainty? The relations you are deriving may be too complex for most applications.

"The joint PDF for polarimetric observations obtained for a single pulse can be found in Middleton (1996, chapter 9.2). Single-pulse measurements, however, are rarely used in the radar meteorology because of the low sensitivity and higher requirement for storage space. The observed radar spectra, almost always, result from the averaging of a number of return pulses. Unfortunately, a solution for the case of averaging over a number of pulses is not yet available in literature."

I am not sure if the authors are completely correct. Please see the reference below, where analytical forms of pdf of estimates dual-pol variables are derived. While the derived pdfs are given for SAR observations, the underlying assumptions are the same.

Jong-Sen Lee, K. W. Hoppel, S. A. Mango and A. R. Miller, "Intensity and phase statistics of multilook polarimetric and interferometric SAR imagery," in IEEE Transactions on Geoscience and Remote Sensing, vol. 32, no. 5, pp. 1017-1028, Sept. 1994, doi: 10.1109/36.312890.

**Line 165:** Please explain what Ns actually means. I expect that you imply that Ns stands for number of independent spectra used to compute an average. This means that spectra are computed from non overlapping time sequencies. Is this correct?

**Section 5.** "Measurements were made during a rain event on 21 June 2021 at 7:44 UTC. I/Q measurements provide high data rate of about 900 MB min−1. Therefore, about 3 min of I/Q measurements were collected for the analysis. Since different chirp types 275 have different properties, in the following only I/Q data collected with the first chirp type are used. Taking into account that the first chirp type has 37 range bins, in total 2.2 × 103 chirp sequences (15.9 × 106 chirps) are available in each polarimetric channel."

A number of questions arises. What did you measure, rain, cloud, etc? What was SNR? What is the impact of noise (I suggest that you state explicitly what is the impact of noise on your formulations)? You need to give a better description of the observations.

---

## Author Comment (AC1)

We would like to thank the reviewer for the comments on our manuscript. Below we give our responses. The original reviewer comments are shown in blue color, while our responses are in black color.

Reviewer: The authors address this problem by constructing a different set of variables (labeled as the vector **b** as opposed to the traditional vector **c**) that allows "for a proper analytic treatment of their error covariance matrix". They also make generalization to the case of the covariance error matrices with nonzero nondiagonal elements. It is shown that the theoretical expressions for the error covariance matrix with respect to the vector **b** are in better agreement with their estimations from the real radar data than the ones with respect to the vector **c** as demonstrated by the comparison of Figs. 8 and 9 in the manuscript.

Response: We would like to emphasize that the vector **b** is not something we invent. The elements of the vector **b** are an intermediate step between raw radar measurements (I/Q data) and the widely known elements of the vector **c** (reflected power, ZDR, Rho_hv, Phi_dp). In other words, in order to calculate elements of the vector **c**, one calculates elements of the vector **b** first. One of our main messages is that it is much easier to properly describe the covariance matrix of the vector **b** than the one of the vector **c**.

Reviewer: I do believe that this study raises an important issue and is a step in the right direction although I personally do not foresee wide utilization of the variational retrieval methods for practical applications in the near future due to their complexity.

Response: We agree with the reviewer that variational retrievals are not yet widely used in meteorological radars. There are however a number of studies we cite in the introduction section, which show first successful attempts. We expect that with increasing computation power these methods are becoming more and more widely used. Also note, that methods which require measurement error covariance matrix are not limited by variational retrievals. Sensitivity analysis and data assimilation are performed by many national weather services, academic institutions and private companies and often lack a proper treatment of the measurements uncertainties. Here as well there are first attempts made to assimilate polarimetric weather radar data. We already declared the range of possible applications of the study at Lines 11, 56, 231, 261, 379.

I do not have any specific comments and suggestions mainly because the manuscript contains plenty of mathematical derivations that are quite difficult to follow and check, and a reader has to trust the authors regarding their validity.

Response: This manuscript is a result of few years of work. We agree that it has a very specific content loaded with mathematical expressions. We, however, see this as a necessity to ease the following of the mathematical derivation of all the analytic formulas. All the steps required to understand how results were derived and explicitly written. Only the most lengthy derivations are pushed to the appendix section. An interested reader familiar with linear

algebra and statistics should be able to follow the explanations. For every step we give references and detailed explanation. In addition, we share the raw radar data and the code, which can be downloaded and checked against the sample data used in the paper or even the original data. Therefore, although we value the reader's trust we think that we made every possible step to overcome any possible doubt towards this work. We think that the statement "a reader has to trust the authors" is unjustified, because everything can be checked. We made the manuscript and supplementary material as opened as possible.

I am not even sure that the paper is a good fit for the AMT journal. It may be more suitable for Journal of the Atmospheric Sciences but this is, of course, for the editor to decide.

Response: We believe that our manuscript perfectly fits the declared scope of the journal, which we quote: "The main subject areas comprise the development, intercomparison, and validation of measurement instruments and techniques of data processing and information retrieval for gases, aerosols, and clouds. **Papers submitted to AMT must contain** atmospheric measurements, laboratory measurements relevant for atmospheric science, and/or **theoretical calculations of measurements simulations with detailed error analysis** including instrument simulations." This is further reinforced by the aims of the AMT special issue "Fusion of radar polarimetry and numerical atmospheric modelling towards an improved understanding of cloud and precipitation processes" which clearly focuses on the practical applications of polarimetric radar measurements. On the other hand, the suggested alternative, Journal of the Atmospheric Sciences (JAS): "publishes basic research related to the physics, dynamics, and chemistry of the atmosphere". Our manuscript is focused on measurement uncertainties and does not contain any results about physics, dynamics, or chemistry of the atmosphere.

---

## Author Comment (AC2)

We would like to thank Prof. Moisseev for the valuable comments. The input definitely helped us to improve the manuscript. Below each of the comments is addressed. The comments are shown in blue color, while our responses are in black.

**Reviewer comment 1:** The main problem of this manuscript, it is not easy to read.

**Response:** Indeed, the manuscript has a lot of information and mathematical expressions, which might be difficult to follow for a reader not familiar with the topic. We, however, did our best to make the manuscript as open as possible. We explain all general steps in the main body of the manuscript leaving the details of the mathematical derivations in the appendix. In addition, we published the data and all scripts required to completely reproduce the results. The scripts can be also applied to different data in order to ease the application of the derived formula to original studies. We believe that an interested reader should be able to prove the results we got. For a less experienced user, the ready-use-scripts are available in the supplement.

**Reviewer comment 2:** In equation (9) you use Bhh, Zdr, rho_hv and Phi_dp. Why do you use Bhh and not Zhh, which would be more commonly used? Of course, Bhh is not Zhh, it is a spectral Zhh. But the same applies to Zdr and the rest of variables. It would be good if you would try to use more widely used notations.

**Response:** Please note, that Bhh is a variable proportional to the power of a spectral line in the horizontal channel. Depending on the exact radar system it can be different quantities. It can be calibrated in reflectivity units (e.g. in RPG cloud radars), but it can be also calibrated in Watts, or even in arbitrary units (e.g. METEK cloud radars). There is no difference in using any of these quantities as Bhh. This is the reason why we do not want to restrict formulas to the spectral reflectivity only. The notation Bhh, Bvv, and Bhv we used to be consistent with the previous study on cloud radar spectral polarimetry (Myagkov et al 2016). We added the following sentence to the manuscript: "Note, that in general $B_{hh}$, $B_{vv}$, and real and imaginary parts of $\dot{B}_{hv}$ can be calibrated in any quantity that is proportional to the power (Watts) received by the radar; e.g. classical radar reflectivity (mm$^6$~m$^{-3}$) or even arbitrary units \citep{Myagkov2015a}."

**Reviewer comment 3:** On line 89, page 3. You state "For each transmitted pulse (the term pulse is used throughout the study, although for radars with frequency modulated continuous wave signals a chirp would have been implied)". Then in Section 5 you return to using chirp and introduce chirp sequence. It took me some effort to adjust to that transition. I would suggest that you either use pulses or chirps.

**Response:** Thanks for pointing this out. We agree that this is misleading. We think that using the term 'pulse' fits better to the introduction section because pulsed radars are more widely spread in the meteorological community. On the other hand, in the validation section we use

an FMCW radar which uses chirps. We decided to add the following to the manuscript: "Since pulsed radars are currently more common in the meteorological community, we use the term "pulse" to refer to a type of the transmitted radar signal in Secs. 2–4. For radars with frequency modulated continuous wave (FMCW) signals, however, the term "chirp" should be used. Later, in the Sec. 5 we use measurements from a FMCW radar and therefore the term "chirp" is used there.

**Reviewer comment 4:** The other concern is whether results of this study will ever be used. Because the formulations are rather complex, there is a good chance that they will never be adopted. Is there an approximation that can be used and that would work for most applications? If yes, could you make a recommendation.

**Response:** Please note, that the aim of the manuscript is two-fold. (1) The given error model is a very important component of our own retrieval algorithms currently being developed. In order to make it possible to refer to the error model instead of describing this complex math in every following manuscript we would like to publish the error model first. (2) We agree that there are simpler models (with approximations and assumptions) available and there are many studies using them (examples are given in the introduction section). We do not see a point in making the long path through the complex math to get the exact solution and afterwards to reduce its accuracy by assumptions and approximations. If an available simplified error model fits reader needs it can be used. We just want to make the exact solution available for the community as well. As mentioned earlier, we also provide ready-to-use matlab scripts in the supplement. The scripts allow a reader to directly use our results without going through the math and without implementing it from a scratch.

**Reviewer comment 5:** Of course, the other option could be a follow up study, demonstrating practical applications.

**Response:** We are currently working on a few variational retrieval techniques using the given error model. It is our plan to publish them in near future. Giving an application example in the current manuscript would make it even larger and even more complex. We added the following to the Summary section: "In order to demonstrate a practical application of the developed characterization of the measurements errors, a few retrieval techniques are being currently developed. The first one is an improvement of the ice-share retrieval described in \citet{Myagkov2015a}. Another one is an adoption of the drop-size-distribution from \citet{Tridon2015} for dual-polarimeteric cloud radar observations."

**Reviewer comment 6:** Abstract: "This study presents the first-ever complete characterization of random errors in dual-polarimetric spectral observations of meteorological targets by cloud radars." While this statement is true, at least to my

knowledge, it seems to me that the underlying assumptions are not very, if at all, different from what are used to describe normal dualpolarization observations. Could you please elaborate what are the main differences? I would argue that the derived expressions presented in this study are representing a subset of conditions for which expressions in (Doviak and Zrnic) or (Chandrasekar and Bringi) are derived. One big simplification, which I believe is valid for spectra observations at least to some extent, used in this study is that averaging is performed using independent spectra, i.e. Ns represents number of independent averages. That means that one does not need to consider the correlation function, which is a must for time-domain observations. I think a more complete discussion on this should be included in the article. You are giving a short discussion on line 105. I would suggest that you compare and contrast the assumptions used in this study and more classical studies.

**Response:** Please note, that the discussion in Lines 97—108 of the original version of the manuscript considers the correlation between neighboring range and spectral bins. This correlation is not related to the correlation of samples in time domain. We give some general considerations about possible sources and implications of the correlation in range and spectral domain but, as stated in the manuscript, detailed analysis of these effects is out of the scope of the manuscript. We think that Prof. Moisseev was referring more to the correlation in the time domain.

Prof. Moisseev is totally right, we do not develop a new approach for the error characterization. In fact, we follow the very same approach from classical works of Doviak, Bringi, Zrnic, and Chandrasekar. And we cite their works a lot throughout the manuscript. The reviewer is absolutely right that in the case of spectra we make a simplification that time samples of a spectral line are not correlated. This is because the coherency period of the scattering is smaller than time required to collect pulses for a single FFT. It can be proved using Eq. 5.2 in Doviak et al 1979 that states that samples are only significantly correlated if:

$\frac{\lambda}{2T} \gg \sigma$, where $\lambda$ is the wavelength, T is sample period, and $\sigma$ is spectral width. Assuming the wavelength of 3.2 mm, T = 1/10000 [Hz] * 256 = 0.0256 s (10000 kHz is a typical pulse repetition frequency, 256 is a typical number of pulses for FFT) the spectral width must be much smaller (at least an order of magnitude) than 0.0625 m/s in order to have some significant coherency. Since in the atmosphere there is almost always a turbulence stronger that few cm/s, it can be safely assumed that there is no coherency between samples of a spectral line.

Using this simplification, we derived an analytical solution for the error characterization applicable to cloud radar spectral polarimetry and this is exactly the novelty of our work. We agree that it is may not be clear enough from the manuscript. To make it clearer we implement the following modifications

(1) explicitly saying that we use exactly the same assumptions as in classical works in the beginning of section 2: "This section introduces relations between a raw cloud radar signal, complex amplitudes, and spectral polarimetric variables for observations of meteorological targets. These relations are based on the same set of assumptions introduced in classical works of Doviak et al. (1979) and Bringi and Chandrasekar (2001) for precipitation radars."

(2) Adding the explanation why we can assume that samples of a spectral line are not coherent. "Unlike precipitation radars which perform rapid azimuth scans, cloud radars are typically pointed to a certain direction or make slow scans to get non-broadened Doppler spectra. \citet{Doviak1979} showed (Eq.~5.2 in there) that the coherency between the adjacent samples depends on the wavelength and the sample repetition period. Cloud radars typically have the pulse repetition frequency in the order of $10$~kHz and $N_\text{ftt}$ in the range from 128 to 1024. This results in getting a single spectrum every 0.01--0.1~s. For such sampling properties of cloud radars any significant coherency between adjacent samples of a spectral line requires the spectral broadening not exceeding at most a few cm~s$^{-1}$. The turbulent spectral broadening, however, exceeds few cm~s$^{-1}$ even in stratiform non-precipitating clouds \citep{Borque2016}. Therefore, consecutive samples of complex amplitudes for a spectral line can be considered to be independent."

(3) adding the following to the section 3: "The estimators Eqs. 15–18 are the same as given in Bringi and Chandrasekar (2001, Chapter 6.4.5). The only difference is that within this work the variables are calculated using complex amplitudes for a spectral line instead of185using I/Q components as is done by precipitation radars"

**Reviewer comment 7:** "One of the main conclusions of the study is that the convenient representation of spectral polarimetric measurements including differential reflectivity ZDR, correlation coefficient ρHV , and differential phase ΦDP is not suited for the proper characterization of the error covariance matrix." I believe you need to explain this point more explicitly. In the summary the discussion is rather short: "It is illustrated that elements of Σc have considerable differences from those estimated from the measurements. The differences are related to the first-order Taylor approximation which does not take into account non-linearities. In contrast, Σb agrees well with the observations. The correlation between calculated elements of Σb with those estimated from the observations exceeds 0.965."

**Response:** We added the following to the summary section: "First, we found differences in variances of $Z_{DR}$, $\rho_{HV}$, $\Phi_{DP}$ of up to factor of 10, 5, and 100, respectively. Second, the calculated variance of $\Phi_{DP}$ shows unrealistically high values by far

exceeding the range of possible values. Third, most of the off-diagonal terms of $\Sigma_c$ are not correlated with corresponding values estimated from observations."

**Reviewer comment 8:** I did not understand if the problem is due to the first-order Taylor approximation used or something else. In many applications an approximation would perfectly well. What would you recommend using to get an estimate of the uncertainty? The relations you are deriving may be too complex for most applications.

**Response:** The statistics of the vector b fits well to the observations. On the other hand, the statistics of the vector c derived by the first-order Taylor from the vector b does not fit to observations. This means that the first order Taylor approximation is the reason. And it is reasonable explanation, because the approximation assumes linear relations between elements of the vector b and the elements of the vector c. But of course, it is clear that these relations are by far not linear. When the signal to noise ratio is high (in precipitation radar community this is often checked by having phv close to 1), then of course variability is quite low and the Taylor approximation gives reasonable results. At low signal-to-noise ratio, however, the variability is high and the non-linearity introduces much larger errors into the variance estimates. The extreme effect can be seen in the variance of PHIDP (see figure 4d), where the variance estimate goes far beyond values PHIDP can take. We added the following to the summary section: "We relate the differences to the first-order Taylor approximation. The Taylor approximation assumes linear relations between elements of the vector b and the elements of the vector c, while the relations include highly non-linear functions."

In the summary sector we have given the recommendation to use the statistics of the vector b instead of c. As explained above, it is reasonable to use the simplified approximations when they give acceptable results. For cases when this is no longer possible (e.g. low SNR) our manuscript offers exact solutions with ready-to-use matlab functions. We added "When the signal to noise ratio is high (> 35 dB), however, the variances are quite low and the Taylor approximation may give reasonable results."

**Reviewer comment 9:** "The joint PDF for polarimetric observations obtained for a single pulse can be found in Middleton (1996, chapter 9.2). Single-pulse measurements, however, are rarely used in the radar meteorology because of the low sensitivity and higher requirement for storage space. The observed radar spectra, almost always, result from the averaging of a number of return pulses. Unfortunately, a solution for the case of averaging over a number of pulses is not yet available in literature." I am not sure if the authors are completely correct. Please see the reference below, where analytical forms of pdf of estimates dual-pol variables are derived. While the derived pdfs are given for SAR observations, the underlying assumptions are the same. Jong-Sen Lee, K. W. Hoppel, S. A. Mango and A. R. Miller, "Intensity and phase statistics of multilook polarimetric and

interferometric SAR imagery," in IEEE Transactions on Geoscience and Remote Sensing, vol. 32, no. 5, pp. 1017-1028, Sept. 1994, doi: 10.1109/36.312890.

**Response:** we are especially grateful for this comment! We were not aware of this publication. We added the following text to the introduction section: "\citet{Lee1994} showed a derivation of a joint probability density function of polarimetric variables for the case of averaging. The authors used a number of assumptions applicable for Earth's surface observations using synthetic-aperture radars. It turns out that the same assumptions are applicable to spectral polarimetric observations of meteorological targets. This allows for using a similar approach in analytic characterization of errors of spectral polarimetric observations."

**Reviewer comment 10:** Line 165: Please explain what Ns actually means. I expect that you imply that Ns stands for number of independent spectra used to compute an average. This means that spectra are computed from non overlapping time sequencies. Is this correct?

**Response:** Yes, this is correct. We clarified this in the text.

**Reviewer comment 11:** What is the impact of noise (I suggest that you state explicitly what is the impact of noise on your formulations)?

**Response:** We agree, that this point is not covered in the manuscript. The noise in both polarimetric channels is not known exactly. Typically, it is estimated from spectra using e.g. the Hildebrand-Sekhon algorithm. A subtraction of noise levels from corresponding diagonal terms of the covariance matrix **B** to get an estimate of signal-only power leads to occasions when the noise corrected matrix **B** is no longer positive semidefinite. In this case the correlation coefficient calculated from noise corrected matrix **B** can exceed 1, which is mathematically speaking should not be possible. In order to avoid this problem, we use the signals as they are measured by the radar i.e. signal + noise. In other words, our used polarimetric variables are not "intrinsic" but affected by noise. Only in this case we can properly describe the statistics. This we explicitly discussed in the manuscript now (end of the section 2.2).

**Reviewer comment 12:** Section 5. "Measurements were made during a rain event on 21 June 2021 at 7:44 UTC. I/Q measurements provide high data rate of about 900 MB min−1. Therefore, about 3 min of I/Q measurements were collected for the analysis. Since different chirp types 275 have different properties, in the following only I/Q data collected with the first chirp type are used. Taking into account that the first chirp type has 37 range bins, in total 2.2 × 103 chirp sequences (15.9 × 106 chirps) are available in each polarimetric channel." A number of questions arises. What did you measure, rain, cloud, etc? What was SNR? You need to give a better description of the observations.

**Response:** In general, it does not matter what kind of meteorological target (rain or cloud) we use for the validation. More important is to make sure that the target has properties of the meteorological target and this we check in Sec. 5.2. But we agree that a better description can be provided, however, in the original manuscript it is written that we collected 3 min of I/Q measurements during a rain event. We extended the measurement description as follows: "The radar was pointed to $45^\circ$ elevation. Since the first chirp sequence covers the lowest part of the atmosphere, the analyzed data correspond to rain. As explained in Sec.~\ref{sec:spec_pol}, no noise subtraction is required to describe the statistics of the measurements. We therefore, use all available spectral lines, including those containing noise only. 90\% of spectral noise power was from 0.2--1.3$\times10^{-3}$~[a.u]. Signal-to-noise ratio (defined here as a ratio of signal power in a spectral line divided by the mean spectral noise power in the same range bin) specified in linear units was from 0 (no signal) to $10^6$. We would like to emphasize, that no filtering based on signal-to-noise ratio was applied."

---

## Editor Decision (ED1)

Dear Alexander, dear Davide,

based on the two reviews received on your manuscript "Analytic characterization of random errors in spectral dual-polarized cloud radar observations" from two experts in the field of radar polarimetry, I plan to give your comments and an updated manuscript free for a second iteration of review. But before I do this, I would like you to consider a further modification of the manuscript.

Reviwer#1 states "…the manuscript contains plenty of mathematical derivations that are quite difficult to follow and check…"

and

Reviewer#2 states "The main problem of this manuscript, it is not easy to read."

I have studied your replies to the reviewers and am convinced that your manuscript contains the complete set of mathematical derivations for an experienced radar user to understand and apply your methodology. However, I think you need to address a broader audience. Radar operators nowadays are not only experts in the field of radar polarimetry, so please make this manuscript also comprehensible for scientists using cloud radar from a variety of applications and even disciplines.

Especially Sections 3 and 4 are dominated by equations and references to the appendices. Please make the scientific narrative clearer, it is there in principle, but not well recognizable. This could be improved by explaining why you are deriving which equation in more detail and discussing the outcomes of your derivation in relation to the original objective. This could also include formulating the title of the sub-sections in a more generally understandable way and relating a discussion of derivations unambiguously to these titles. Also, please consider more text details between the steps of your equation derivations. I'm sure that this will make your manuscript more accessible to a broader range of interested scientists.

I am looking forward to receiving your modified manuscript.

Best regards

Uli

---

## Author Response (AR2)

**Editor's comment:**

Reviwer#1 states "...the manuscript contains plenty of mathematical derivations that are quite difficult to follow and check..." and Reviewer#2 states "The main problem of this manuscript, it is not easy to read." I have studied your replies to the reviewers and am convinced that your manuscript contains the complete set of mathematical derivations for an experienced radar user to understand and apply your methodology. However, I think you need to address a broader audience. Radar operators nowadays are not only experts in the field of radar polarimetry, so please make this manuscript also comprehensible for scientists using cloud radar from a variety of applications and even disciplines. Especially Sections 3 and 4 are dominated by equations and references to the appendices. Please make the scientific narrative clearer, it is there in principle, but not well recognizable. This could be improved by explaining why you are deriving which equation in more detail and discussing the outcomes of your derivation in relation to the original objective. This could also include formulating the title of the sub-sections in a more generally understandable way and relating a discussion of derivations unambiguously to these titles. Also, please consider more text details between the steps of your equation derivations. I'm sure that this will make your manuscript more accessible to a broader range of interested scientists.

**Response:**

First, we would like to thank the Editor for the comment. We agree that the manuscript in its current state could be difficult to follow for a reader not familiar with the topic. We therefore reworked the manuscript considerably. The introduced changes are listed below:

1. We added two sentences into the introduction briefly explaining what has been done within the study.

2. According to the Editor's recommendation, we split the section 2 into a number of small subsections focused on certain topics reflected in their titles.

3. We extended the introduction of the section 3. We explain in more details the steps taken in this section. We also added a reference where the same procedure is used.

4. We modified titles of subsections in the section 3 and added sentences guiding the reader through results and following steps.

5. We added the subsection 3.4 to better split the results derived for different basses.

6. In the section 3.4. we now explicitly state the final result of the section.

7. In the section 4 we extended the introduction to better guide a reader from the section 3 to 4.

8. In subsections 4.1 and 4.2 we added more sentences clearly stating the starting point and final results

9. In the subsection 5.5. we added a sentence to emphasize that the first element of the vectors b and c are the same.

10. In the section 6 we made it clear that no additional processing is required to get the vector b since it is an intermediate step from IQ data to conventionally used representation including Bhh,ZDR,rho,PHI.

We believe that the introduced modifications indeed improve the readability of the manuscript. If, however, the manuscript is still considered as too complicated, we highly appreciate if a list of specific points which are difficult to follow is provided.